# Multi-phosphorylation reaction and clustering tune Pom1 gradient mid-cell levels according to cell size

Veneta Gerganova[1], Charlotte Floderer[2], Anna Archetti[2], Laetitia Michon[1], Lina Carlini[2], Thais Reichler[1], Suliana Manley[2]*, Sophie G Martin[1]*

[1]Department of Fundamental Microbiology, Faculty of Biology and Medicine, University of Lausanne, Lausanne, Switzerland; [2]Institute of Physics, School of Basic Science, École Polytechnique Fédérale de Lausanne (EPFL), Lausanne, Switzerland

**Abstract** Protein concentration gradients pattern developing organisms and single cells. In *Schizosaccharomyces pombe* rod-shaped cells, Pom1 kinase forms gradients with maxima at cell poles. Pom1 controls the timing of mitotic entry by inhibiting Cdr2, which forms stable membrane-associated nodes at mid-cell. Pom1 gradients rely on membrane association regulated by a phosphorylation-dephosphorylation cycle and lateral diffusion modulated by clustering. Using quantitative PALM imaging, we find individual Pom1 molecules bind the membrane too transiently to diffuse from pole to mid-cell. Instead, we propose they exchange within longer lived clusters forming the functional gradient unit. An allelic series blocking auto-phosphorylation shows that multi-phosphorylation shapes and buffers the gradient to control mid-cell levels, which represent the critical Cdr2-regulating pool. TIRF imaging of this cortical pool demonstrates more Pom1 overlaps with Cdr2 in short than long cells, consistent with Pom1 inhibition of Cdr2 decreasing with cell growth. Thus, the gradients modulate Pom1 mid-cell levels according to cell size.
DOI: https://doi.org/10.7554/eLife.45983.001

*For correspondence:
suliana.manley@epfl.ch (SM);
Sophie.Martin@unil.ch (SGM)

**Competing interests:** The authors declare that no competing interests exist.

## Introduction

In many organisms and cell types, graded protein patterns provide positional information. This is true from the smallest bacteria, where polar gradients of protein activity define the position of the division apparatus (*Kretschmer and Schwille, 2016*), to the largest multicellular organisms, where morphogen concentration gradients define regions of gene expression during development (*Briscoe and Small, 2015*). Although mechanisms of gradient formation vary, in all systems the graded patterns are thought to convey information at a distance from the source.

In fission yeast, concentration gradients formed by the DYRK-family kinase Pom1 have received considerable attention, due to the role of Pom1 in regulating the timing of mitotic entry and thus cell size at division (*Martin and Berthelot-Grosjean, 2009*; *Moseley et al., 2009*). Pom1 gradients are nucleated at cell poles upon dephosphorylation by a type I phosphatase complex whose regulatory subunit Tea4 is delivered by microtubules (*Hachet et al., 2011*; *Martin et al., 2005*; *Tatebe et al., 2005*). Dephosphorylation of Pom1 reveals a lipid-binding activity that maps to a 200-aa region in its N-terminal half. At the plasma membrane, Pom1 forms small clusters and is thought to laterally diffuse (*Hachet et al., 2011*; *Saunders et al., 2012*). It also undergoes autophosphorylation reactions that promote its detachment from the membrane, leading to the graded pattern (*Hachet et al., 2011*).

An interesting feature of Pom1 gradients is a noise correction mechanism that compensates for large variations in protein concentration at the cell poles, which can vary up to fourfold (*Hersch et al., 2015*; *Saunders et al., 2012*). In a simple diffusive gradient, the decay length (the

**eLife digest** All organisms need to know how to arrange different cell types during the development of their organs and tissues. This information is provided by protein concentration patterns, or gradients, that tell cells how to behave based on where they are positioned. The same fundamental principles also work on a smaller scale. For example, although the rod-shaped yeast *Schizosaccharomyces pombe* is a single-celled organism, it uses protein concentration gradients to control its growth and timing of division.

Before *S. pombe* cells divide, they need to check that they have reached the right size. Several mechanisms contribute to this information. One of them involves a concentration gradient of a protein known as Pom1, which is found on the cell membrane, with more protein at the cell extremities and less towards the middle. Pom1 serves to block the activity of Cdr2 – an enzyme that localizes to the cell middle and controls cell division. An open question has been whether Pom1 levels at the center drop as the cell grows, coordinating growth and division.

One explanation for how the Pom1 gradient could be regulated is by the removal and addition of phosphate groups. At the cell's tip, an enzyme removes phosphate groups from Pom1, causing it to bind to the membrane. As Pom1 diffuses along the membrane, it continuously 're-phosphorylates' itself. This promotes Pom1 to gradually detach, restricting it from spreading along the membrane towards the cell middle. Another explanation is that clusters of Pom1, formed at the membrane, help establish a gradient by moving along the membrane at different rates: larger clusters, formed in high concentration areas, move slower than smaller clusters, causing levels of Pom1 to be higher at the tip, and lower towards the middle. Now, Gerganova et al. set out to find which of these two processes contributes more to shaping the Pom1 gradient, and determine where Pom1 acts on Cdr2.

Gerganova et al. used super resolution microscopy to track individual Pom1 molecules inside yeast cells. This revealed two findings. First, that individual Pom1 molecules do not travel all the way from the cell tip to the center, but 'hop' between clusters as they move towards the middle. Second, in longer cells levels of Pom1 on the membrane drop at the center, where Pom1 encounters Cdr2. As a result, Cdr2 will come across higher levels of Pom1 in short cells, but low levels of Pom1 in long cells. This allows Pom1 to act as a measure of cell size, preventing short cells from dividing too soon.

The role of clusters in creating gradients is not only relevant for yeast cell division. It could potentially apply to the gradients that organize cells and tissues in different organisms. Future work could examine whether similar principles apply in more complex systems.

DOI: https://doi.org/10.7554/eLife.45983.002

distance at which the concentration is reduced to a certain fraction of its amplitude) is independent of the amplitude at the pole. In contrast, the Pom1 gradient is corrected by varying the slope of the gradient decay: gradients with higher Pom1 concentration have a steeper decay, while those with lower Pom1 concentration have a flatter decay. Two models have been proposed to explain the source of this correction. One model suggests gradient buffering is the consequence of concentration-dependent inter-molecular phosphorylation, which promotes Pom1 detachment from the membrane (*Hersch et al., 2015*). This model also explains the correction for the even larger variations in levels of Tea4 concentration at cell poles. A second model hypothesizes that buffering results from concentration-dependent clustering of Pom1, with higher concentrations leading to larger, slower diffusive clusters, causing a traffic-jam effect at the cell tips (*Saunders et al., 2012*). However, direct experimental evidence testing these models is scarce.

Pom1 has two physiological functions. First, Pom1 provides spatial information for division site positioning: *pom1Δ* cells divide off-center (*Bähler and Nurse, 2001*; *Bähler and Pringle, 1998*). Second, Pom1 provides temporal information to regulate the timing of mitotic entry: *pom1Δ* cells divide precociously at an aberrantly small size (*Martin and Berthelot-Grosjean, 2009*; *Moseley et al., 2009*). For both functions, Pom1 directly phosphorylates the SAD-family kinase Cdr2, but on different residues (*Bhatia et al., 2014*; *Rincon et al., 2014*). Pom1 function in division site placement likely also involves additional substrates. To delay mitotic commitment, Pom1 phosphorylates the

C-terminus of Cdr2 (*Bhatia et al., 2014*; *Deng et al., 2014*), which antagonizes the activating phosphorylation of the Cdr2 kinase domain by the cytosolic CaMKK Ssp1 (*Deng et al., 2014*). Cdr2 localizes at the mid-cell cortex, where it forms large, stable clusters called nodes, which contain many other proteins including a second SAD-family kinase Cdr1 (*Martin and Berthelot-Grosjean, 2009*; *Morrell et al., 2004*; *Moseley et al., 2009*). The signal relay between Cdr2 and Cdr1 is not yet elucidated, but the output is an inhibitory phosphorylation of Wee1 kinase, which itself exerts direct inhibitory activity on the sole cyclin-dependent kinase CDK1 (*Kanoh and Russell, 1998*; *Young and Fantes, 1987*). In contrast to the stable Cdr1 and Cdr2 association to the nodes, Wee1 visits are only transient (*Allard et al., 2018*; *Moseley et al., 2009*). These visits increase in frequency and duration as cells grow, consistent with the idea that Wee1 is inactivated in longer cells to permit CDK1 activation and mitotic entry.

Although genetic and biochemical evidence have firmly established Cdr2 as Pom1 substrate, there has been much debate on where the Pom1-Cdr2 interaction takes place, and whether the strength of this interaction varies in the course of a single cell cycle. Initial work proposed Pom1 gradients as a means to measure cell size, because total fluorescence measurements of Pom1-GFP along cell length revealed higher medial fluorescence in short than long cells (*Martin and Berthelot-Grosjean, 2009*; *Moseley et al., 2009*). This led to the model that Pom1 inhibits Cdr2 in short cells, but that inhibition is relieved upon attaining sufficient cell size, thus coupling cell size with mitotic entry. Two lines of evidence indicate that Pom1 activity on Cdr2 indeed varies with cell size. First, the levels of Cdr2 phosphorylation by Ssp1 increase during G2, consistent with a progressive decrease in Pom1-dependent inhibition (*Deng et al., 2014*). Second, the frequency and duration of Wee1 visits to Cdr2 nodes increase as cells grow, with direct evidence showing that Pom1 suppresses Wee1 visits in short cells (*Allard et al., 2018*). In addition, Pom1 re-localization to cell sides, which is prominent upon glucose starvation, leads to strong mitotic delay (*Kelkar and Martin, 2015*). However, subsequent analyses of cortical fluorescence profiles on confocal mid-plane images failed to detect significant differences in the levels of Pom1-GFP at mid-cell between short and long cells, raising questions about the previously proposed model (*Bhatia et al., 2014*; *Pan et al., 2014*). The apparently invariant Pom1 mid-cell levels in cells of various lengths led to the suggestion that Pom1 may control Cdr2 activity elsewhere (*Bhatia et al., 2014*), or is not involved in cell size homeostasis (*Pan et al., 2014*), in agreement with the observation that *pom1Δ* cells retain homeostatic capacity (*Wood and Nurse, 2013*). Because the number of Cdr2 nodes at mid-cell increases with cell surface growth, this also led to the suggestion that Cdr2, not Pom1, may be the critical cell size sensor in the pathway (*Pan et al., 2014*). Thus, there is currently a controversy between the invariant Pom1 levels at mid-cell and the size-dependent effect of Pom1 on Cdr2 function.

In this work, we have coupled generation of a systematic *pom1* mutant allelic series with a wide range of imaging methods – including single molecule super-resolution PALM (photo-activated localization microscopy) imaging and tracking, confocal and TIRF (total internal reflection fluorescence) microscopy – to obtain quantitative information on the patterning of Pom1 gradients. This enabled three major findings: first, Pom1 gradients are primarily shaped by phosphorylation-mediated detachment with clusters acting as the relevant membrane-associated unit; second, Pom1 regulates Cdr2 for mitotic entry at the mid-cell cortex; third, TIRF imaging reveals significantly higher levels of Pom1 at the mid-cell cortex in short cells.

## Results

### Membrane diffusion and dissociation of Pom1

We investigated the diffusion and membrane dissociation dynamics of single molecules of Pom1 in *S. pombe* cells. Cells were prepared either on a flat agarose pad and imaged horizontally along their long axis, or on a micropatterned surface imprinted with holes, where cells oriented vertically for cell pole imaging (*Figure 1A*, *Figure 1—figure supplement 1*). We relied on photoconversion, localization, and tracking of single fluorescent proteins in living cells, in our case the Pom1-mEos3.2 fusion expressed from the native genomic locus. Single molecules of Pom1-mEos3.2 were observed along the side or pole of the cell, with a higher density of localizations at the poles consistent with the known Pom1 density gradient, and tracked by monitoring their position over time to produce single molecule trajectories (*Figure 1B*) (*Manley et al., 2008*). We used astigmatic imaging to ensure that

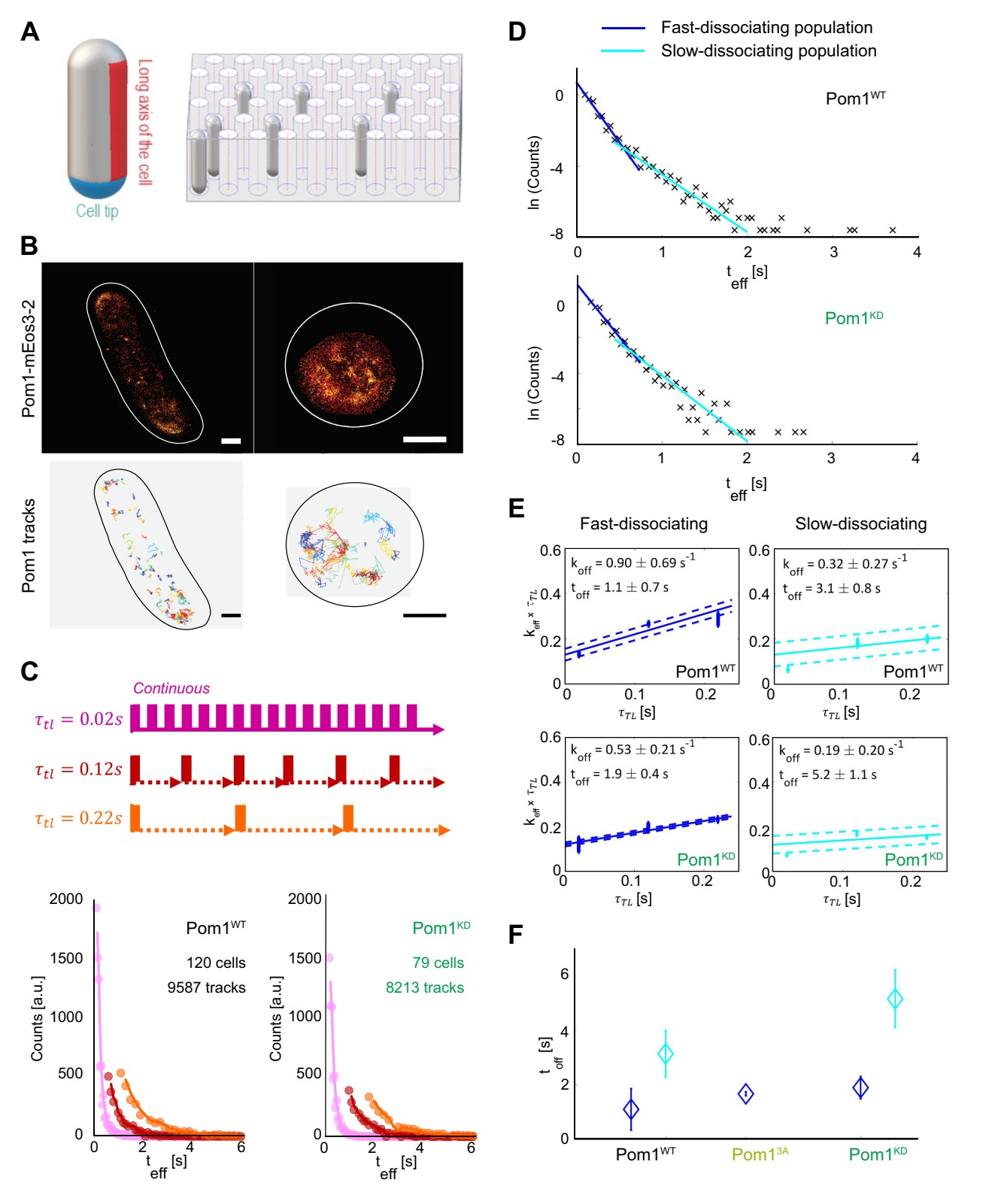

**Figure 1.** Pom1 dissociation dynamics. (**A**) (left) Scheme of the cell regions imaged on flat pads (cell side, red) or vertical molds (cell pole, blue). (right) Scheme of the mold used for vertical immobilization in agar. (**B**) PALM reconstruction of Pom1 on the cell side (top left) or the cell pole (top right). White lines correspond to cell boundaries. Corresponding sptPALM tracks on the side (bottom left) and on the pole (bottom right). Scale bar 1 μm. (**C**) Scheme for time-lapse imaging experiments. Every frame is recorded with a 20ms exposure time (solid rectangles). A time delay (dashed arrows) is

*Figure 1 continued on next page*

*Figure 1 continued*

introduced between each pair of consecutive frames. The time-lapse period ($\tau_{TL}$) is the sum of the integration time and the time delay. Effective residence time distributions for Pom1$^{WT}$ and Pom1$^{KD}$, color correspond to different $\tau_{TL}$. (D) Residence time distributions for Pom1$^{WT}$ and Pom1$^{KD}$, fit with a bi-exponential decay: fast (blue) and slow (cyan). (E) Effective rate constant as a function of time lag condition (symbols) for fast (blue) and slow (cyan) populations. Solid lines correspond to a weighted linear fit according to *Equation 2*. Error bars correspond to the weights associated to each data point (S.D. from the fit of the exponential distribution of the residence time obtained according to *Equation 1* in Methods). (F) Comparison of the Pom1 residence time obtained for each subpopulation (fast in blue and slow in cyan) for Pom1$^{WT}$, Pom1$^{3A}$ and Pom1$^{KD}$ strains. Error bar corresponds to the standard deviation of the parameter extracted from the weighted linear fit of panel 1E.

DOI: https://doi.org/10.7554/eLife.45983.003

The following figure supplements are available for figure 1:

**Figure supplement 1.** Fabrication of micro-holes for *S. pombe* cell vertical immobilization.

DOI: https://doi.org/10.7554/eLife.45983.004

**Figure supplement 2.** Comparison of Pom1-GFP and Pom1-mEos3.2.

DOI: https://doi.org/10.7554/eLife.45983.005

**Figure supplement 3.** Pom1 dissociation analysis.

DOI: https://doi.org/10.7554/eLife.45983.006

we tracked only the molecules in the plane of the membrane (see Materials and methods). The Pom1-mEos3.2 distribution along cortical profiles was indistinguishable from that of Pom1-GFP (*Figure 1—figure supplement 2*).

To determine the dissociation rate of single molecules, we performed time-lapse imaging on horizontally oriented cells with several lag times $\tau_{TL}$ (*Gebhardt et al., 2013*) (*Figure 1C*). This allows the effects of dissociation and photobleaching to be analytically separated, since varying the lag time will vary the contribution of photobleaching and therefore the effective residence time ($t_{eff}$) (*Figure 1C*), while the actual residence time of molecules remains unchanged. Interestingly, we observed that the residence time exhibits a multimodal distribution, which could be fitted with a bi-exponential decay corresponding to short and long residence times, or fast and slow dissociation rates (*Figure 1D*, *Figure 1—figure supplement 3A*). The majority of Pom1 molecules (76%) comprise the fast-dissociating part, while the remaining (23%) dissociate more slowly. This last population contained few molecules, but its long tail could be further explained by the presence of two or more slowly dissociating sub-populations.

To understand the contribution of Pom1 auto-phosphorylation in membrane detachment, we analyzed the time-lapse data of Pom1 binding using both Pom1$^{WT}$-mEos3.2 and kinase-dead Pom1$^{KD}$-mEos3.2 to extract binding times ($t_{off}$) and dissociation rates ($k_{off} = 1/t_{off}$) (*Figure 1E*). Both Pom1$^{WT}$ and Pom1$^{KD}$ showed fast and slow-dissociating populations, with a similar fold difference in dissociation rates between the two populations (2.8x and 2.7x, respectively). Interestingly, when we compared the fast-dissociating populations in the two strains, the Pom1$^{WT}$ population dissociated 1.7x faster than the Pom1$^{KD}$ population ($t_{off}^{fast}$ = 1.1 ± 0.7 s and 1.9 ± 0.4 s for Pom1$^{WT}$ and Pom1$^{KD}$, respectively). The slowly dissociating population showed a similar trend ($t_{off}^{slow}$ = 3.1 ± 0.8 s and 5.2 ± 1.1 s for Pom1$^{WT}$ and Pom1$^{KD}$, respectively) (*Figure 1E-F*). We note that an even slower sub-population may exist, shown as the tail distribution in *Figure 1D*, but this represents a very small fraction of Pom1 molecules (<1%) for which we lack sufficient number of tracks to extract a reliable dissociation rate.

Thus, Pom1 activity, which leads to auto-phosphorylation, promotes a faster dissociation rate of Pom1 from the membrane, in agreement with previous biochemical observations (*Hachet et al., 2011*). Furthermore, both Pom1$^{WT}$ and Pom1$^{KD}$ dissociation kinetics exhibit at least two distinct populations perhaps corresponding to different multimerization states.

## Localization and diffusion of Pom1

To study the lateral diffusion of Pom1 at the plasma membrane, single fluorescent Pom1 proteins were tracked (*Figure 1B*) and analyzed to extract their diffusion coefficients. We found that the track duration $t_{eff}$ and diffusion coefficient $D_{eff}$ were inversely correlated (*Figure 2A*), with shorter tracks exhibiting faster diffusion and longer tracks exhibiting slower diffusion. Since long residence times imply a slower dissociation time, slowly diffusing Pom1 molecules present slow dissociation

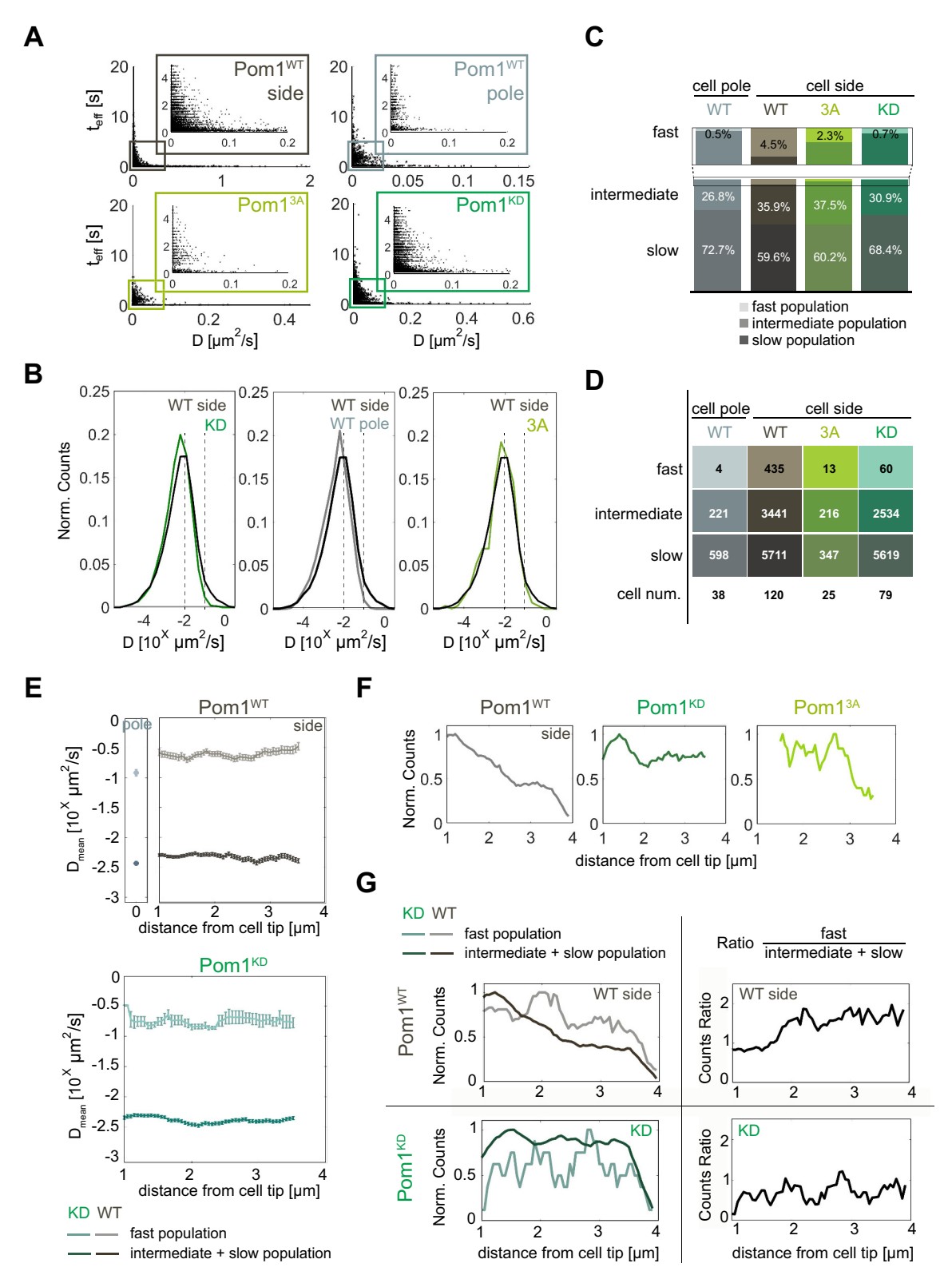

**Figure 2.** Pom1 diffusion dynamics. (A) Track length as a function of diffusion coefficients of the tracks for Pom1[WT] (WT) at cell sides (dark grey) and cell poles (light grey), Pom1[3A] (3A; light green), Pom1[KD] (KD; green-cyan). Pom1[3A] and Pom1[KD] were imaged at cell sides. (B) Distribution of diffusion coefficients of all Pom1 molecules tracked in Pom1[WT], Pom1[3A] and Pom1[KD] at cell sides, and Pom1[WT] at cell poles. Thresholds used in panels C-G are shown by the dashed lines.(C) Proportion of fast (D $\geq 10^{-1}$ μm²/s; light), intermediate ($10^{-2} \leq$ D $< 10^{-1}$ μm²/s; medium color) and slow (D $< 10^{-2}$ μm²/

*Figure 2 continued on next page*

*Figure 2 continued*

s; dark) populations for Pom1^WT, Pom1^3A and Pom1^KD at cell sides, and Pom1^WT at cell poles. (**D**) Number of tracks and cells for each condition. (**E**) Average diffusion coefficient as a function of the distance from the pole for fast ($D \geq 0.1$ μm²/s; light color) and combined intermediate and slow (dark) populations for Pom1^WT ($D < 0.1$ μm²/s; top panels) and Pom1^KD (bottom) strains. Error bar corresponds to standard error of the mean. (**F**) Evolution of count of all tracks along the cell length normalized by the maximum occurrence for Pom1^WT, Pom1^KD and Pom1^3A. (**G**) Evolution of the number of tracks for fast ($D \geq 0.1$ μm²/s; light color) and combined intermediate and slow ($D < 0.1$ μm²/s; dark) populations along the cell length for Pom1^WT and Pom1^KD strains (left panels) and their ratio (right panel) for Pom1^KD and Pom1^KD strains. The color coding dark grey = Pom1^WT at cell sides, light grey = Pom1^WT at cell poles, green-cyan = Pom1^KD, and light green = Pom1^3A is used throughout the figure.

DOI: https://doi.org/10.7554/eLife.45983.007

The following figure supplements are available for figure 2:

**Figure supplement 1.** Pom1 diffusion coefficient analysis with different thresholds.

DOI: https://doi.org/10.7554/eLife.45983.008

**Figure supplement 2.** Pom1^3A diffusion coefficient analysis with different thresholds.

DOI: https://doi.org/10.7554/eLife.45983.009

dynamics, and vice versa. This suggests that in larger, slowly diffusing clusters, Pom1 remains more stably associated with the membrane.

Pom1^WT and Pom1^KD exhibit a broad distribution of diffusion coefficients (*Figure 2B*) consistent with previous measurements by fluorescence correlation spectroscopy (*Saunders et al., 2012*). The distribution of Pom1^WT at cell sides was shifted toward slightly faster diffusion compared with Pom1^KD. To better understand this difference, we defined two thresholds at $D \geq 10^{-1}$ μm²/s and $D \geq 10^{-2}$ μm²/s, which separate molecules into three sub-populations of fast ($D \geq 10^{-1}$ μm²/s), intermediate ($10^{-2} \leq D < 10^{-1}$ μm²/s) and slow-diffusing molecules ($D < 10^{-2}$ μm²/s) (*Figure 2B*). Note that in this analysis, all fast-diffusing molecules are also fast-dissociating, but intermediate and slow-diffusing molecules can exhibit either fast- or slow-dissociating behaviors. Interestingly, there was a substantially higher proportion of fast-diffusing molecules for Pom1^WT than Pom1^KD, which also on average diffused faster ($D_{mean} = 0.31 \pm 0.01$ μm²/s for Pom1^WT and $0.21 \pm 0.02$ μm²/s Pom1^KD; *Figure 2C–E*). We note that the average diffusion rate of these three populations did not vary along cell length with distance from cell poles (*Figure 2E*). These analyses were robust to changes in threshold choice (*Figure 2—figure supplement 1*). Thus, the fast population of Pom1^WT diffuses faster than Pom1^KD.

We then performed the same analysis on Pom1^WT at the cell pole using cells oriented vertically (*Figure 1A*). Interestingly, the distribution of diffusion coefficients was shifted toward slower values compared to Pom1^WT at the cell sides (*Figure 2B*). Indeed, when using the same thresholds as above, a much smaller proportion of molecules were fast-diffusing compared with the sides (*Figure 2C*; *Figure 2—figure supplement 1A–C*), very similar to the proportion observed for Pom1^KD. This is consistent with Pom1 being mainly in its dephosphorylated state at the cell pole, in agreement with the presence of the Tea4-PP1 phosphatase at this location (*Hachet et al., 2011*).

We then tracked the evolution of the number of events along cell length starting from the cell tip. The overall number of events decreased for Pom1^WT but not Pom1^KD, consistent with their described localization patterns (*Figure 2F*). Interestingly, when we considered fast, intermediate and slow populations separately, we found that Pom1^WT exhibits a change in the relative proportion of molecules: the proportion of fast diffusive Pom1 remained relatively constant all along the gradient, but the proportion of intermediate and slow diffusive Pom1 decreased, resulting in a relative increase in the fast population (*Figure 2G*, *Figure 2—figure supplement 1E–G*). In contrast, the proportions of fast, intermediate and slow diffusive populations of Pom1^KD remain balanced all along the gradient, leading to a nearly constant ratio.

In summary, these measurements provide two important insights: First, the measured binding times and diffusion rates indicate that individual Pom1 molecules cover on average a small distance before detaching from the membrane. Let's consider the fast-diffusing molecules. These represent 4.5% of the population, diffuse on average at 0.31 μm²/s and are all part of the 76% of molecules binding the membrane for an average time of 1.1 s. In fact, given the inverse correlation observed between diffusion and binding times, they likely bind the membrane for an even shorter time. From these values, we can estimate a maximum travelled distance of 0.8 μm. Slower-diffusing molecules travel an even shorter distance before detaching. Thus, it is unlikely that individual molecules

continuously track from cell pole to cell sides. Second, the shorter binding time of fast-diffusive molecules and their progressive increase in proportion at a distance from the cell pole for Pom1$^{WT}$ but not Pom1$^{KD}$ suggests the possibility this may be caused by progressive phosphorylation-dependent Pom1 detachment from larger clusters. This led us to examine more closely the mode of Pom1 attachment to the membrane and the role auto-phosphorylation plays to shape the gradient.

## Pom1 binds the plasma membrane through two distinct motifs

Previous work has shown that Pom1 interacts with lipids (*Hachet et al., 2011*) and a fragment containing amino acids 305 to 510 can efficiently bind the plasma membrane (*Figure 3A*, fragment #1). A BH-search prediction (*Brzeska et al., 2010*) performed for this fragment identified two potential membrane binding sites (*Figure 3B*). These two regions map to the most conserved sequences in the 305–510 fragment: the first (aa 437–444) is within a 22aa-long sequence (MB1; aa 423–444) that is identical in the four *Schizosaccharomyces* species (*S. pombe, S. octosporus, S. japonicus, and S. cryophilus*), the second (aa 480–492) falls within a weak amphipathic helix prediction (MB2; aa 477–494) (*Figure 3A–C*). To test the validity of these predictions, we constructed a series of shorter and/or mutagenized GFP-tagged Pom1 fragments integrated as single copy in wildtype and *pom1Δ* cells. The localization of all fragments tested was identical in wildtype and *pom1Δ* cells (*Figure 3D* and *Figure 3—figure supplement 1*).

The Pom1 fragment spanning aa 305–510 localizes uniformly at the plasma membrane (fragment #1). Truncation of the C-terminal 20 aa (fragment 305-490aa, fragment #2), which cuts into the predicted amphipathic helix, resulted in a strong reduction (though not complete loss; see below) of Pom1 cortical localization. Progressive N-terminal truncations showed that a minimal 468-510aa fragment (fragment #5) containing the putative amphipathic helix was sufficient for membrane binding. In this fragment, converting the hydrophobic Ile494 to the polar, uncharged Asp residue disrupted membrane binding (I494N, fragment #6), indicating that Pom1 binds the membrane through the predicted amphipathic helix. However, the same point mutation in the full 305-510aa fragment (fragment #7) did not disrupt cortical localization, indicating the presence of a second lipid-binding domain. We then mutagenized seven amino acids within the conserved MB1 region to alanine (generating the mutant allele MB1*, fragment #8), which also on its own did not perturb membrane binding. However, combining both the I494N and MB1* mutations within fragment 305–510 yielded a fully cytosolic localization (fragment #9). We conclude that Pom1 localization to the plasma membrane relies on two adjacent lipid-binding motifs.

To confirm the results from the fragment analysis, we introduced the same mutations in full-length Pom1 at the native genomic locus. Individual I494N and MB1* mutations led to a decrease of fluorescence intensity at the cell tip, which was exacerbated in the double mutant. Nevertheless, in this double mutant small Pom1 clusters were visible at the cell tips, likely due to Pom1 direct binding to the phosphatase regulatory subunit Tea4 through PxxP motifs (*Hachet et al., 2011*). Indeed, additional mutagenesis of the five previously identified PxxP motifs in combination with the MB1* and I494N mutations rendered Pom1 entirely cytosolic.

## A Pom1 allelic series shows additive features of multi-phosphorylation

Pom1 autophosphorylation promotes membrane detachment. From over 40 phosphorylation sites identified in silico and by mass-spectrometry analysis, combined mutation of six of these sites was previously shown to abolish the Pom1 gradient ([*Hachet et al., 2011*]; note that each site contains up to three serines or threonines mutated in aggregate). Five of these sites are located in the 305-510aa fragment: numbers 1, 2, and 3 are within the conserved MB1 membrane-binding region, while 4 and 5 are located in the MB2 amphipathic helix (*Figure 3A*, sites indicated by black boxes). The sixth one is distal to the kinase domain and was not directly investigated here.

To test the contribution of multi-phosphorylation for gradient shape and buffering, we generated a series of endogenously tagged phospho-blocking Pom1 alleles, carrying alanine substitution in 1, 2, 3 or 5 phosphorylation sites (*Figure 4A*, top row), and quantified Pom1 gradients by measuring the fluorescence profile at the cell cortex in medial plane confocal images. A decrease in the number of phosphorylation sites led to a gradual flattening of the gradient shape manifested in a decrease of Pom1 intensity at the cell tip and an increase at the cell middle (*Figure 4B–C*, left panels). The gradual change of gradient shape indicates the additive nature of multiple autophosphorylation

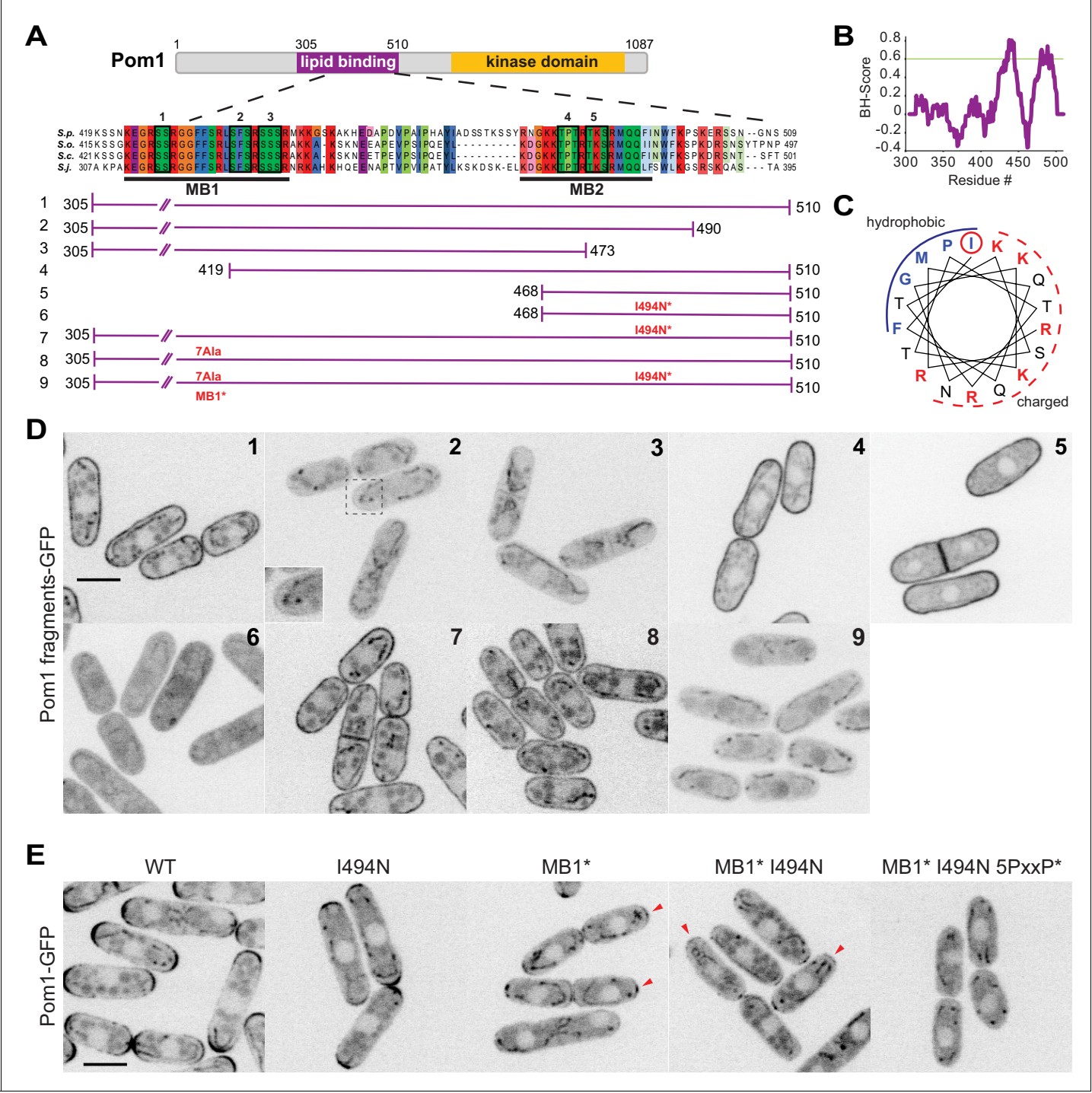

**Figure 3.** Two distinct regions define Pom1 membrane binding. (A) Schematic representation of Pom1 with sequence homology alignment of the lipid-binding region shown for *S. pombe, S. octosporus, S. cryophilus,* and *S. japonicus.* Phosphorylation sites are indicated by black boxes and numbered. The conserved membrane binding region MB1 (aa423-444) and the amphipathic helix membrane binding region MB2 (aa477-494) are underlined. A schematic representation of the fragments used in the truncation analysis presented in D is shown below. (B) BH-search prediction performed on Pom1 lipid-binding region (aa305-510) shows two peaks with a BH-score above 0.6, corresponding to membrane binding regions MB1 and MB2. (C) Predicted amphipathic helix for MB2 with marked residue I494, targeted for mutagenesis I494N. (D) Localization of GFP-tagged fragments 1 to 9 presented in A in wild-type cells. Inset shows residual membrane localization for fragment #2. Scale bar 5 μm. (E) Localization of full-length Pom1WT-GFP and indicated mutants expressed at the native locus. Mutagenesis of the two membrane-binding regions and the 5 PxxP sites that mediate direct binding to Tea4 (*Hachet et al., 2011*) renders Pom1 cytosolic. Red arrowheads indicate residual cell tip localization in Pom1MB1* and Pom1MB1*-I494N mutants. Scale bar 5 μm.

*Figure 3 continued on next page*

Figure 3 continued

DOI: https://doi.org/10.7554/eLife.45983.010

The following figure supplement is available for figure 3:

**Figure supplement 1.** Analysis of Pom1 fragment localization in pom1Δ cells.

DOI: https://doi.org/10.7554/eLife.45983.011

events and reveals that no particular site contributes to Pom1 gradient shape more than another. The gradient shape of the Pom1[5A] mutant was indistinguishable from those of the previously described non-phosphorylatable Pom1[6A] and inactive Pom1[KD] (*Hachet et al., 2011*), indicating that these five sites represent the principal sites modulating Pom1 localization.

To further probe the influence of phosphorylation on Pom1 membrane binding, we used single molecule time-lapse imaging of Pom1[3A]-mEos3.2 to extract dissociation and diffusion rates. The fast-detaching population of Pom1[3A] molecules showed membrane binding times intermediate between Pom1[WT] and Pom1[KD] ($t_{off}^{fast}$ = 1.7 ± 0.1 s; *Figure 1*, *Figure 1—figure supplement 3B*). The number of tracks was not sufficient to determine the residence time of any slower-dissociating population with confidence. Thus, consistent with the fact that only some of the auto-phosphorylation sites are mutated in this allele, Pom1[3A] molecules bind the membrane longer than Pom1[WT], but not as long as Pom1[KD]. The distribution of diffusion coefficients was also intermediate between Pom1[WT] and Pom1[KD] (*Figure 2*). The thresholds defined above (as in *Figure 2B*) similarly showed a lower proportion of fast-diffusing molecules than Pom1[WT] but a higher proportion than Pom1[KD] (*Figure 2C-D*, *Figure 2—figure supplement 1A-C* and *Figure 2—figure supplement 2*). These data are consistent with auto-phosphorylation promoting Pom1 detachment from the plasma membrane.

To assess phosphorylation-dependent gradient shape changes in a simplified system containing a single membrane-binding site, we used the Pom1[MB1*] mutant. This mutant binds the membrane solely with its amphipathic helix, which contains only phospho-sites 4 and 5. Consistent with poorer membrane binding, Pom1[MB1*] gradient profiles showed decreased intensity compared to wildtype, at both cell poles and cell sides (*Figure 4D*). Mutagenesis of phospho-site 5, generating Pom1[MB1*-1A], led to further gradient flattening with decreased Pom1 intensity at cell tips and increase at cell middle (*Figure 4D*). Thus, autophosphorylation regulates each of the two membrane-binding sites.

The progressive increase in medial cortical Pom1 levels in the phospho-site allelic series is consistent with the previously proposed idea that sequential phosphorylation events provide a timer function for Pom1 diffusion from cell poles (*Hachet et al., 2011*). Additionally, non-phosphorylated Pom1 alleles may directly bind the membrane at the cell sides. To test the second scenario, we monitored the localization of the allelic series in tea4Δ cells, which lacks the phosphatase regulatory subunit (*Alvarez-Tabarés et al., 2007*; *Hachet et al., 2011*). In tea4Δ cells, all Pom1 phospho-mutants bound the cortex nearly uniformly (*Figure 4A*, bottom row), with levels that increased with the number of phospho-site mutations (*Figure 4B–C*, right panels), and a concomitant decrease in Pom1 cytosolic levels (*Figure 4—figure supplement 1C*). Again, Pom1[5A] cortical levels were indistinguishable from Pom1[6A], but a little lower then Pom1[KD], for unknown reasons. These data are in agreement with the idea that each phosphorylation event progressively lowers membrane affinity to reduce Pom1 binding in the medial region. We note that changes in Pom1 distribution in the phospho-site mutants are not due to a change in Pom1 protein concentration, as verified by western blot analysis (*Figure 4—figure supplement 1B*). Comparing Pom1 medial cortical levels in WT and tea4Δ cells showed significantly lower amounts for Pom1 and Pom1[1A] in tea4Δ, but higher or similar levels for Pom1[2A], Pom1[3A] and Pom1[5A]. This suggests that the flatter gradients observed in the Pom1 phospho-mutants are not only due to a reduction of Pom1 detachment from the membrane, allowing lateral diffusion over a longer distance, but also to an increase in Pom1 attachment to the membrane at mid-cell. Thus, Pom1 auto-phosphorylation both favors its detachment from, and prevents its attachment to, the membrane.

## Pom1 phospho-mutants have reduced gradient shape robustness

A key feature of the Pom1 gradient is its robustness to variations within the system. Previous work showed that variability in Pom1 concentration at cell poles is counteracted by varying the gradient decay length, which leads to a strong negative correlation between decay length and Pom1

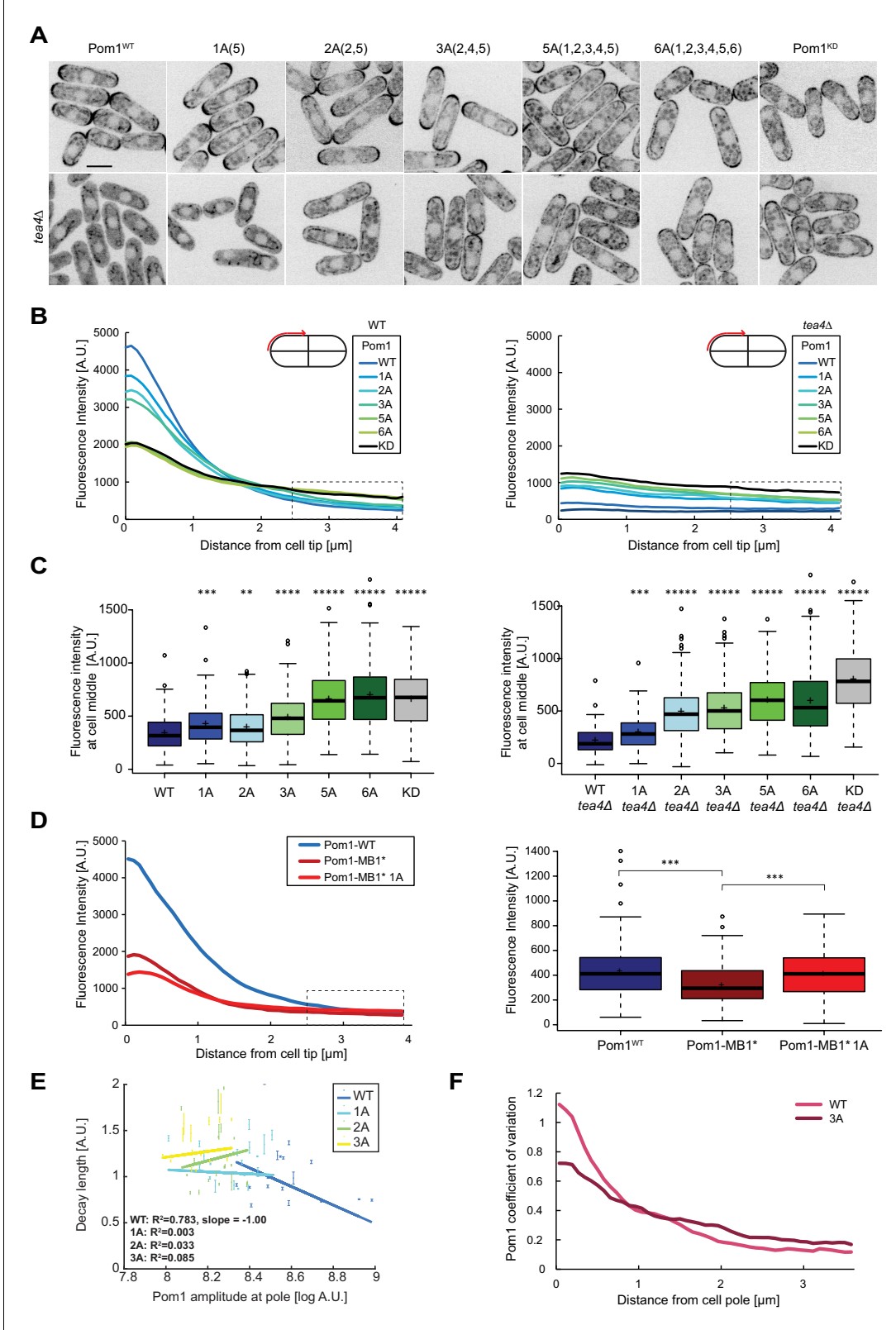

**Figure 4.** Pom1 gradient shape and robustness depend on multisite auto-phosphorylation. (**A**) Medial plane confocal images of a series of Pom1 phospho-blocking alleles in otherwise wild-type (top row) and *tea4Δ* cells (bottom row). Scale bar 5 μm. (**B**) Fluorescence intensity plots of cortical gradient profiles, collected from time-average medial plane confocal images as shown on the schematic. Left: wild type background; right: *tea4Δ* cells. Graphs show averages of 240 gradient profiles per strain, n = 3 experiments, 20 cells per experiment. Individual experiments are shown in *Figure 4—*

*Figure 4 continued on next page*

*Figure 4 continued*

**figure supplement 1**. The dashed box shows the region selected for Pom1 intensity measurements at mid-cell shown in panel C. (**C**) Mean Pom1 fluorescence intensity levels at cell middle, extracted from the last 1.5 μm of profiles shown in panel B. Left: wild-type background, right: *tea4Δ* cells. (**D**) Cortical gradient profiles of Pom1$^{WT}$, Pom1$^{MB1*}$ and Pom1$^{MB1*-1A(5)}$ (left) and corresponding quantification of Pom1 intensity at cell middle (right), as in panels B and C. Graphs show averages of 160 gradient profiles per strain, n = 2 experiments, 20 cells per experiment. (**E**) Decay length plotted against Pom1 amplitude at the cell pole. Each dot represents an average gradient profile from bin sorting of 5%. Error bars correspond to the percentage of fitting quality for each dot. (**F**) Coefficients of variation for Pom1$^{WT}$ and Pom1$^{3A}$ from the 240 gradient profiles shown in B. Means indicated by plus sign, error bars: SD, statistical significance measured against wild type unless otherwise indicated by t-test with unequal variances. \*\*, p=0.0001, \*\*\*p=$10^{-8}$, \*\*\*\*p≤$10^{-18}$, \*\*\*\*\*p≤$10^{-40}$.

DOI: https://doi.org/10.7554/eLife.45983.012

The following figure supplement is available for figure 4:

**Figure supplement 1.** Analysis of Pom1 phospho-site mutant alleles.

DOI: https://doi.org/10.7554/eLife.45983.013

amplitude at the pole (*Hersch et al., 2015*; *Saunders et al., 2012*). Pom1 clustering and Pom1 inter-molecular phosphorylation have both been theoretically proposed as source for this correction, though we so far lack experimental evidence. We assessed the contribution of the multi-phosphorylation reaction in gradient buffering by plotting the correlation between decay length and amplitude at the pole in Pom1 phospho-mutants that retain significant pole enrichment. Pom1$^{1A}$, Pom1$^{2A}$ and Pom1$^{3A}$ all showed a loss of negative correlation of decay length to Pom1 concentration at the cell tip (*Figure 4E*). Thus, these Pom1 mutants poorly correct intrinsic variations of Pom1 concentrations at the cell tips. Another evidence for buffering comes from the steep decrease in the coefficient of variation of Pom1$^{WT}$ from the cell tip to the cell middle (*Hersch et al., 2015*). While we confirm this observation with our current data set, the decrease in the coefficient of variation at cell tips and cell sides is much smaller for Pom1$^{1A}$, Pom1$^{2A}$ and Pom1$^{3A}$ (from 9.6-fold for Pom1$^{WT}$ to 6.6 for Pom1$^{1A}$, 5.8 for Pom1$^{2A}$, and 4.2 for Pom1$^{3A}$; *Figure 4F*, *Figure 4—figure supplement 1D*). We conclude that the Pom1 phosphorylation cycle directly contributes to Pom1 gradient shape robustness.

## Pom1 at the mid-cell cortex controls cell length at division

One important physiological role of Pom1 is to set cell size at division by negatively regulating the SAD-family kinase Cdr2, which forms stable cortical nodes at mid-cell (*Martin and Berthelot-Grosjean, 2009*; *Moseley et al., 2009*). Consistent with previous reports that Pom1$^{6A}$ and Pom1 overexpression lead to cell size increase at division (*Hachet et al., 2011*; *Martin and Berthelot-Grosjean, 2009*; *Moseley et al., 2009*), we observed a gradual increase in cell length at division in mutants of the phospho-site allelic series (*Figure 5A*). Cell length at division was strongly correlated with the medial cortical Pom1 levels (*Figure 5B*), consistent with the idea that medial Pom1 levels set the cell division size. To evaluate the relative contributions of cytosolic and cortical Pom1 in Cdr2 inhibition, independently of gradient formation, we obtained the same measurements in *tea4Δ* cells: the progressive increase in cortical Pom1 in Pom1 phospho-mutants also correlated with an increase of cell length in *tea4Δ* background (*Figure 5A–B*). We note that *tea4Δ* cell lengths were slightly longer than WT for *pom1* alleles containing up to three phosphosite mutations. By contrast, the cytosolic Pom1 signal in the allelic series was largely invariant in WT and showed a modest decrease in *tea4Δ* leading to an inverse correlation with cell length at division (*Figure 5—figure supplement 1A*). The correlations between cell length and Pom1 levels at cell poles ran in opposite directions in WT and *tea4Δ* cells (*Figure 5—figure supplement 1B*). We conclude that the medial cortical pool of Pom1 is the relevant pool for cell size regulation.

Because Pom1 clusters rather than individual molecules may shape the gradient (see *Figure 1* above), and inspired by recent work showing visits of Cdr2 nodes by the downstream Wee1 kinase (*Allard et al., 2018*; *Gerganova and Martin, 2018*), we turned to live cell TIRF imaging. To test the method's sensitivity and selectivity for cortical signals, we first compared the fluorescence levels of Pom1$^{WT}$, Pom1$^{1A}$, Pom1$^{3A}$ and Pom1$^{5A}$ phospho-mutants, which reproduced the increased Pom1$^{1A}$, Pom1$^{3A}$ and Pom1$^{5A}$ medial cortical localization seen by confocal microscopy (*Figure 5C*). Specific measurements of cluster size and intensity also showed higher values for Pom1$^{3A}$ and Pom1$^{5A}$ (*Figure 5—figure supplement 2*). We then imaged two cytosolic proteins. First, cytosolic GFP,

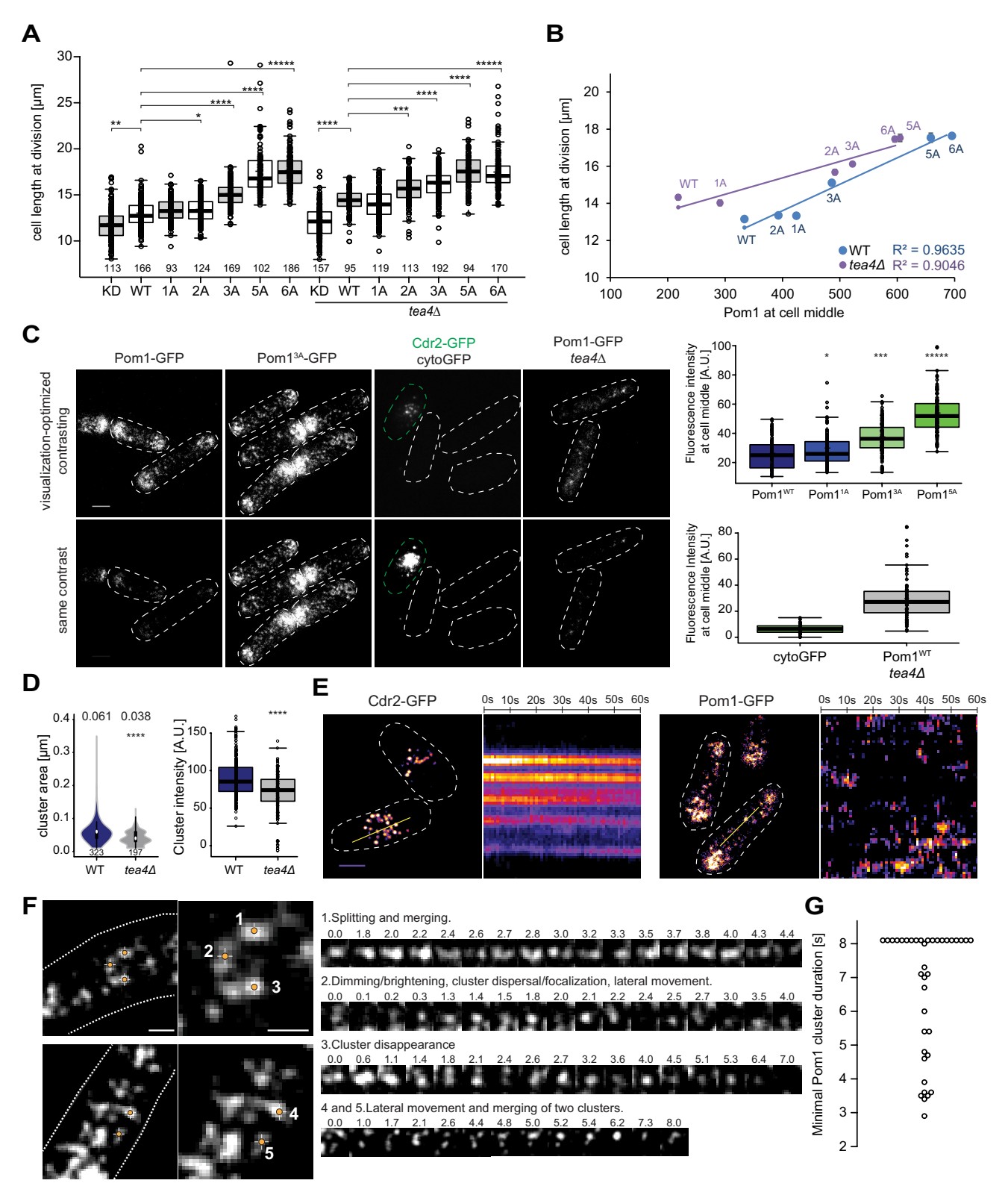

**Figure 5.** Cortical Pom1 at mid-cell regulates cell size. (A) Mean cell length at division of the *pom1* phospho-mutant allelic series in otherwise wild type and *tea4Δ* background with number of quantified cells indicated, *, p=10$^{-2}$, **, p=10$^{-7}$, ***, p=10$^{-10}$, ****, p≤10$^{-20}$, *****, p=10$^{-40}$. (B) Correlation plot of cell length at division (values from panel A) versus Pom1 intensity at cell middle (values from *Figure 4C*). Error bars are standard error. (C) TIRF imaging of Pom1, shown with identical acquisition and contrasting parameters (bottom) and adjusted contrast (top). Quantification of intensities at cell

*Figure 5 continued on next page*

*Figure 5 continued*

middle is shown on the right (strains imaged the same day plotted together). Pom1$^{1A}$-GFP, Pom1$^{3A}$-GFP and Pom1$^{5A}$-GFP show increased cortical levels compared to Pom1$^{WT}$-GFP (p=0.01, p=10$^{-14}$, p=10$^{-41}$, respectively). Note that cytoGFP expressed from the *pom1* promoter strain was mixed with a Cdr2-GFP strain to identify the TIRF focal plane (green dotted line). Scale bar 2.5 µm. Images for Pom1$^{1A}$-GFP and Pom1$^{5A}$-GFP are in supplement 2. (D) Mean cluster area (left) and average cluster intensity (right) for individual Pom1 clusters from wild type versus *tea4Δ* background, n = 25 cells from two individual experiments. Number of clusters indicated below violin plot. The mean area is shown at the top, ****, p=10$^{-15}$. (E) Localization of Cdr2-GFP and Pom1-GFP by TIRF microscopy. Left panels show a snapshot of time point 0 from a time series imaged every second for 60 s. Right panels show kymographs along the dotted line. Scale bar 2.5 µm. (F) Examples of Pom1-GFP cluster behaviors in TIRF, taken from two cells imaged every 100 ms over 8 s. Line 1: Cluster splitting (3.0 s and 4.4 s) and merging (3.2 s). Line 2: Cluster fluorescence fluctuations (down at 0.1 s, 2.0 s, 2.4 s; up at 0.2 s, 2.1 s, 2.5 s). This cluster also exhibits clear lateral movement. Line 3: Cluster fluorescence fluctuation (down at 2.4 s and 3.6 s; up at 2.6 s and 4.0 s) and disappearance (7.0 s). Line 4: Merging of two clusters (at 6.2 s). The bottom cluster exhibits clear lateral movement. Scale bars: 1 µm. (G) Minimal lifetimes of Pom1 clusters in seconds. Note that the lifetime of all but two clusters is underestimated, as they either existed at the start or end of the timelapse, or both, n = 38 clusters from 12 cells.

DOI: https://doi.org/10.7554/eLife.45983.014

The following figure supplements are available for figure 5:

**Figure supplement 1.** Pom1 levels in the cytosol and at cell poles do not correlate with cell length at division.

DOI: https://doi.org/10.7554/eLife.45983.015

**Figure supplement 2.** TIRF imaging of Pom1$^{1A}$, Pom1$^{3A}$ and Pom1$^{5A}$.

DOI: https://doi.org/10.7554/eLife.45983.016

expressed under the *pom1* promoter, could not be detected by TIRF though it was seen by epifluorescence, confirming that the evanescent field detects only cortical molecules (*Figure 5C*, Figure 5—figure supplement 1C). By contrast, Pom1-GFP in *tea4Δ* cells, which by confocal microscopy is not detected at the cortex (see *Figure 4A*; *Hachet et al., 2011*; *Kokkoris et al., 2014*; *Padte et al., 2006*), revealed a cortical signal by TIRF imaging (*Figure 5C*). In *tea4Δ*, Pom1 formed cortical clusters, though cluster number, size and intensity were lower than in wild type (*Figure 5D*), which likely explains why the cortical Pom1$^{WT}$ signal in *tea4Δ* is virtually indistinguishable from the cytosolic signal by confocal microscopy. Thus, very transient encounters of Pom1 at Cdr2 nodes at the cortex, rather than a fully cytosolic Pom1, could account for the longer size of *tea4Δ* cells noted above. We conclude that TIRF provides a highly specific and sensitive imaging setup for Pom1 clusters at the yeast cortex.

In TIRF timelapse imaging, Cdr2-GFP formed stable nodes at mid-cell, which did not move substantially over 60 s (*Allard et al., 2018*), whereas Pom1-GFP clusters were highly dynamic (*Figure 5E*). Fast 100 ms acquisition intervals were used to monitor Pom1 clusters, which exhibited an array of dynamics with clusters moving laterally, splitting or merging. *Figure 5F* and *Videos 1–2* provide representative examples. Example one shows an initial large cluster that splits in two at 3.0 s, remerges, and splits again at 4.4 s. Clusters 4 and 5 provide a second example of a small cluster moving laterally and merging with a larger one. Another common cluster dynamic is exemplified by clusters exhibiting dimming and/or dispersal of signal, followed by an immediate increase in fluorescence. This behavior can be seen in example 2 between times 0.0 s and 0.3 s, 1.8 s and 2.1 s, and 2.2 s and 2.5 s, the latter one dimming to only a few detectable fluorescent pixels. Note that the detection threshold was set at a value higher than the fluorescence levels measured in a strain not expressing GFP (see Materials and methods). Similarly, example three dims and regains fluorescence between 2.1 s and 2.6 s, and 3.2 s and 4.0 s. It then completely disappears at 7.0 s. There were also many instances where the Pom1 signal was too fluid to unambiguously follow individual clusters over time. These fluctuations in the fluorescence signal of individual clusters indicate that clusters often recombine and can gain and loose individual Pom1 molecules over time. Similar dynamics with oscillatory intensities were observed in the Pom1$^{3A}$ and Pom1$^{5A}$ mutants (*Videos 3* and *4*).

To estimate the lifetime of individual clusters, we followed 38 Pom1 clusters in 12 cells. The cluster lifetime from appearance to disappearance or splitting of the cluster ranged to over 8 s, the length of the imaging timeframe (*Figure 5G*). This value is very likely to be under-estimated as the longest-lived clusters were present from start to end of the time-lapse imaging. These values are higher than previous measurements obtained by confocal microscopy (*Saunders et al., 2012*), probably because of the higher sensitivity of TIRF imaging. They are also substantially longer than the binding time obtained by PALM imaging for individual molecules (around 1 to 3 s, see *Figure 1*).

These observations further support the idea that individual Pom1 molecules turn over within single clusters. Therefore, we propose that Pom1 clusters are the functional units shaping the gradient, as their longer residence time would permit diffusion from the cell pole all the way to the zone of action at mid-cell.

## Pom1 overlaps more with Cdr2 in short than long cells

To investigate the Pom1-Cdr2 interaction at the cortex, we acquired Pom1 TIRF images at 1 s interval for 60 s and took snapshot TIRF images of Cdr2 at the start and end of the imaging period in a dual tagged Pom1-GFP Cdr2-tdTomato strain (*Figure 6A*). The Cdr2 snapshots provided the location of nodes to which we mapped individual regions of interest (ROIs), in which we quantified Pom1 intensity in the GFP channel. Indeed, we were able to observe Pom1 encounters of the Cdr2 nodes of various duration within the 60 s imaging period (*Figure 6A–B*). To distinguish whether these encounters are targeted visits or due to random collisions of Pom1 clusters, we shifted the same size ROIs away from, but in the immediate vicinity of, Cdr2 nodes. We observed a very similar pattern for Pom1 in the non-node-associated ROIs, which suggests that laterally moving Pom1 clusters randomly collide into Cdr2 nodes (*Figure 6C*),

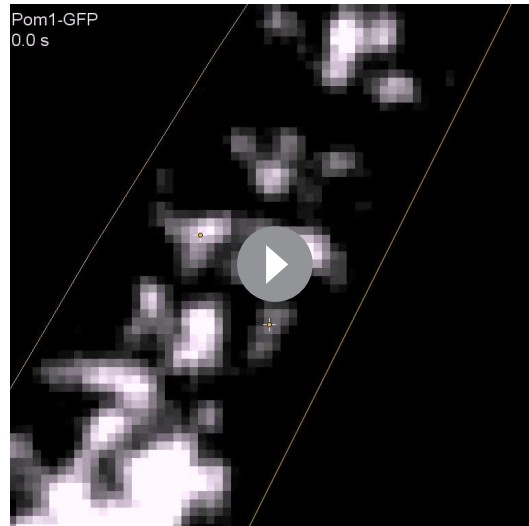

**Video 1.** Examples of Pom1-GFP cluster dynamics. Pom1-GFP was imaged in TIRF mode every 100 ms over a 8 s imaging period. 4 clusters of interest are marked with a target with three prominent clusters visible at the start of the timelapse, and one smaller cluster at the bottom of the frame which appears at 0.2 s and disappears temporarily at 1.2 s and 1.8 s and completely at 5.6 s (possibly as it leaves the evanescent field on the side of the cell). The top positioned cluster splits at 2.8 s (or even sooner), remerges at 3.3 s, splits partially at 3.7 s and completely at 4.4 s, leaving behind two smaller clusters that move laterally and last until the end of the movie. The bottom cluster on at timepoint 0 is observed clearly until 6.9 s. During this period, it dims and brightens, disperses and focalizes several times. At 6.9 s disappears from the field of view, but may not be completely absent as a cluster (not marked) reappears at the same location at 7.0 s. The cluster on the left moves laterally and splits in two clusters at 2.2 s and re-merges at 2.7 s and remains trackable until the end of the 8 s imaging period. All clusters tracked in this movie show examples of dimming, followed by brightening of the signal. The movie is shown in quasi-real time.

DOI: https://doi.org/10.7554/eLife.45983.017

**Video 2.** Examples of Pom1-GFP cluster dynamics. Pom1-GFP was imaged in TIRF mode every 100 ms over a 8-s imaging period. Three clusters of interest are marked with a target. The cluster on the left- splits in three smaller clusters at 0.1 s, after which the clusters appear to detach. The cluster on the top right splits in two clusters at 0.1 s and re-merges briefly at 0.2 s, after which the cluster increases in brightness and splits again at 0.6 s. The small cluster on the left moves laterally, dims in fluorescence and eventually detaches at 1.3 s. The right top cluster and the bottom cluster, marked from the beginning of the movie move laterally, occasionally splitting and merging. The top cluster splits at 2.4 s and re-merges at 2.7 s, as does the bottom cluster at 3.2 s and 3.4 s, and 4.5 s. Eventually, the clusters will merge at 4.9 s, forming a bright and stable cluster, which remains trackable until the end of the 8-s imaging period. The movie is shown in quasi-real time.

DOI: https://doi.org/10.7554/eLife.45983.018

distinct from the targeted visits from the cytosol reported for Wee1 (*Allard et al., 2018*).

Remarkably, when we clustered the data according to cell length, the average Pom1 intensity at all measured Cdr2 nodes was significantly higher in short (6 to 8 μm) than long (12 to 14 μm) cells in three independent experimental repeats (*Figure 6D*, *Figure 6—figure supplement 1A–B*). The values for cells of intermediate length (9 to 11 μm) were more variable, probably depending on the average length of these cells. We observed a similar pattern in the duration of Pom1-Cdr2 encounters, which we defined as the length of time the Pom1-GFP value at a Cdr2 node remained above a defined fluorescence threshold: In short cells, a higher proportion of Cdr2 clusters were continuously occupied by Pom1 over the 60-s imaging period (*Figure 6E*, *Figure 6—figure supplement 1C*). Consistent with the observation that Pom1 behavior is similar at and in the vicinity of Cdr2 nodes, measuring the total medial Pom1 TIRF signal gave similar results: Pom1 levels were higher in the middle of short than long cells (*Figure 6F–G*). Thus, the medial cortical Pom1 levels, which are those critical for Cdr2 regulation, decrease as cells grow, consistent with cell length-dependent relief of Pom1 inhibition.

## Discussion

Investigations on Pom1 concentration gradients have provided two main lines of discussion, one concerning the mode of gradient buffering and another concerning the function of the gradient as a measure of cell length to control the timing of mitotic commitment. In this work we provide evidence for unified models on these two subjects. We confirm experimentally that the buffering of the Pom1 concentration gradient relies on its multi-phosphorylation reaction and establish that Pom1 mole-

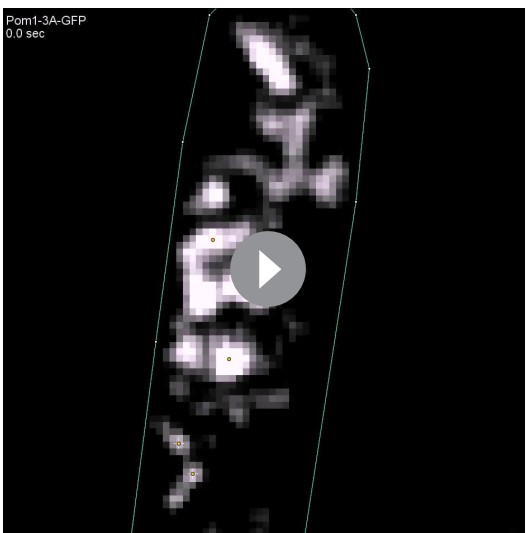

**Video 3.** Examples of Pom1[3A]-GFP cluster dynamics. Pom1[3A]-GFP was imaged in TIRF mode every 100 ms over an 8-s imaging period. Four clusters of interest are marked with a target. The two clusters on the bottom left are short-lived. The top cluster dims at 0.7 s, increases in brightness at 1.1 s and disappears at 1.7 s. The neighbouring small clusters disappear at 0.3 s and 0.9 s. The top two clusters are longer lived with the top cluster dimming at 7.1 s and increasing in brightness at 7.3 s, remaining trackable until the end of the 8-s imaging period, while the lower cluster displays lateral movements between 5.8 s to 6.3 s, dims at 6.5 s and disappears at 6.6 s. The movie is shown in quasi-real time.
DOI: https://doi.org/10.7554/eLife.45983.019

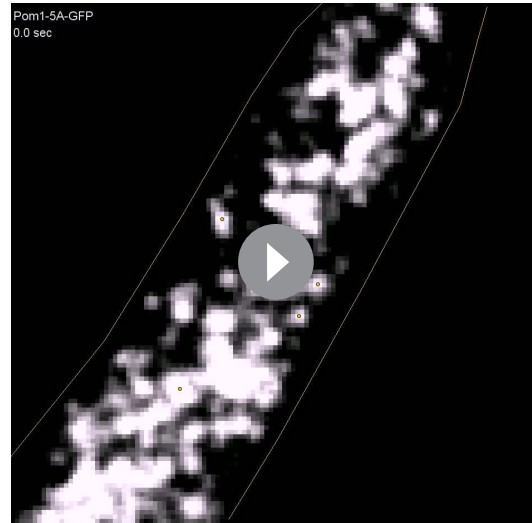

**Video 4.** Examples of Pom1[5A]-GFP cluster dynamics. Pom1[5A]-GFP was imaged in TIRF mode every 100 ms over an 8-s imaging period. Three main areas of interest are marked with targets. The top cluster displays lateral movements between 2.5 s and 3.1 s and disappears at 3.3 s. The middle area tracks three smaller highly dynamic clusters, displaying merging (2.7 s and 4.3 s) and splitting events (3.0 s). The fused cluster shows increase of brightness at 4.4 s and disappears at 7.0 s. The bottom cluster is trackable throughout the imaging period and shows dimming and increase of brightness between 5.8 and 5.9 s. The movie is shown in quasi-real time.
DOI: https://doi.org/10.7554/eLife.45983.020

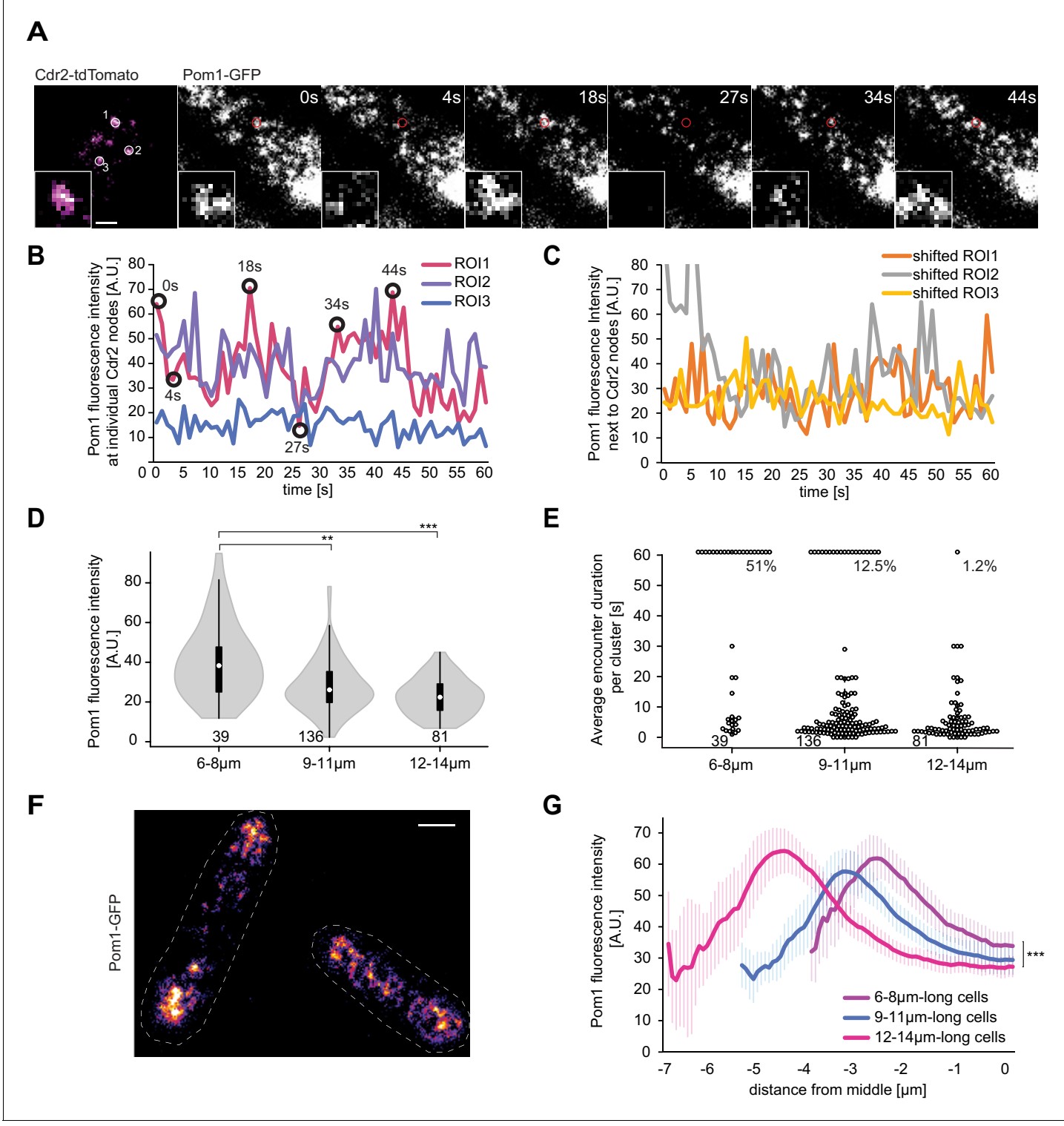

**Figure 6.** Pom1 levels at Cdr2 nodes are higher in short than long cells. (**A**) TIRF images of Cdr2-tdTomato and Pom1-GFP in a dual tagged strain. ROIs (1, 2 and 3) were selected around Cdr2 nodes and fluorescence intensity was measured in the GFP channel (ROI one marked in red). Scale bar 1 μm. (**B**) Pom1-GFP fluorescence intensity in the three ROIs marked in panel A. Circled time points correspond to snapshots in panel A. (**C**) Pom1-GFP fluorescence intensity in ROIs shifted to the immediate vicinity of the ROIs marked in A. (**D**) Average fluorescence intensity of Pom1-GFP at Cdr2 nodes measured as in (A–B) over a 60-s imaging period, sorted by cell length. ***, p≤$10^{-7}$. (**E**) Duration of individual Pom1 encounters with a Cdr2 node for cells of sorted length. This is defined as the length of time the Pom1-GFP signal is over 20 arbitrary fluorescence units. In longer cells, the proportion of clusters with continuous (60 s) Pom1 presence decreases, while the number of shorter encounters increases. For (D–E), data was collected from a single

*Figure 6 continued on next page*

*Figure 6 continued*

experiment: 39 clusters from 6 6–8 µm long cells, 136 clusters from 21 9–11 µm long cells, and 81 clusters from 13 12–14 µm long cells. Experiment duplicates presented in *Figure 6—figure supplement 1*. (F) Localization of Pom1-GFP in TIRF in cells of different lengths (10 µm and 13.5 µm). Scale bar 2.5 µm. (G) Global cortical Pom1 levels measured from TIRF imaging. The gradients are aligned to the cell geometric middle and sorted by cell length. Error bars: standard error between experiments (N = 3); in total 62 gradients from 6 to 8 µm cells, 248 gradients from 9 to 11 µm cells and 128 gradients from 12 to 14 µm cells. Statistics on Pom1 medial levels performed for the last 1.5 µm of the gradient tail between short (6–8 µm) and intermediate cells (9–11 µm), $p=10^{-3}$, and between short and long cells (12–14 µm), $p=10^{-6}$.
DOI: https://doi.org/10.7554/eLife.45983.021
The following figure supplement is available for figure 6:

**Figure supplement 1.** Replicate experiments showing higher Pom1-GFP fluorescence at Cdr2 nodes of short than long cells.
DOI: https://doi.org/10.7554/eLife.45983.022

cules turn over within membrane-bound clusters. The claim of Pom1 gradients serving as a measure of cell length was originally tempered by further investigations that did not detect differences in Pom1 levels at mid-cell. By specifically imaging the relevant pool of Pom1 at the mid-cell cortex, we now unequivocally show that Pom1 gradients formed by diffusion of Pom1 clusters reach the middle of short but not long cells.

## Pom1 clusters as diffusion unit to shape gradients

The Pom1 gradient forms upon local dephosphorylation of Pom1 at the cell pole, promoting Pom1 membrane binding (*Hachet et al., 2011*). Previous modeling work demonstrated that the multi-phosphorylation reaction Pom1 undergoes prior to its membrane detachment could provide a timer function for Pom1 lateral diffusion and serve as a buffering mechanism for the gradient (*Hersch et al., 2015*). An alternative model proposed that differential diffusion coefficients of Pom1 clusters of varying sizes underlie the buffering of the gradient with the largest clusters causing a 'traffic jam' event at the cell tip, which is relieved by the progressive fractionation and the subsequent increase in diffusion coefficients of smaller clusters away from the gradient source (*Saunders et al., 2012*). The data presented here integrates these two mechanisms in a single model for Pom1 gradient formation. Single molecule measurements by PALM imaging show that the gradient is comprised of molecules with a wide distribution of diffusion coefficients. Their dissociation dynamics also revealed at least two populations with distinct binding times. Note that the fast-dissociating population is not identical to the fast-diffusing one. However, there is an overall inverse correlation between binding times and diffusion rates, such that fast-diffusing molecules are also binding the membrane only for a short time. These may represent individual Pom1 molecules not associated with a cluster or in small clusters, and are a minority of all Pom1 molecules. The slower-diffusing molecules may be part of larger clusters. Many of these molecules detach from the membrane slower, but a large pool also exhibits fast dissociation, perhaps indicating peripheral association with the cluster. Importantly, the diffusion coefficients and binding times reveal that individual molecules (in a cluster or not) will only travel a maximum distance of 0.8 µm before detaching from the plasma membrane. Thus, a Pom1 molecule binding at the cell tip upon dephosphorylation by the Tea4-PP1 complex travels only a short distance and does not reach the cell middle.

By contrast, Pom1 clusters as a whole are longer lived at the plasma membrane. Clusters appeared very dynamic, changing intensity over time, splitting, dispersing and fusing again. Importantly, TIRF measurements easily identified cluster lifetimes of over 8 s. This longer lifetime of clusters at the plasma membrane indicates that individual molecules exchange within a cluster. Specifically, this suggests that existing clusters must be able to bind Pom1 molecules directly from the cytosol. In TIRF imaging, we indeed found examples of isolated clusters whose fluorescence intensity diminished before increasing again, providing support for this idea. These longer lifetimes are consistent with clusters being able to form at the cell pole and diffuse laterally all the way to mid-cell, at least in short 7–8 µm cells. Thus, we propose that Pom1 clusters are the functional units that shape the graded distribution of Pom1.

## Pom1 phosphorylation shapes the gradients

The phospho-site mutant allelic series shows that Pom1 distribution critically depends on auto-phosphorylation. Indeed, progressive alanine substitution at up to five auto-phosphorylation sites causes

progressive flattening of the Pom1-graded distribution, with apparently additive contribution of each phosphorylation event. Because the five phosphorylation sites all map very close to each other within two adjacent membrane-binding motifs, it is likely that they have to be phosphorylated sequentially, extending the time frame between the dephosphorylation event taking place at the cell pole and the full auto-phosphorylation, promoting membrane detachment.

Each phosphorylation event reduces the affinity to the plasma membrane, likely affecting both $k_{off}$ and $k_{on}$. Indeed, our PALM data shows that the $k_{off}$ of single Pom1 molecules is modulated by their phosphorylation status: Pom1$^{KD}$, which is not phosphorylated binds the plasma membrane longer than the partly dephosphorylated Pom1$^{3A}$, which itself binds longer than Pom1$^{WT}$. Similarly, alanine-substitutions of phospho-sites led to a progressive increase in the membrane-associated Pom1 fraction in tea4Δ cells, with a fold-change larger than the one measured for the $k_{off}$. This suggests that, although not directly measured, the $k_{on}$ of Pom1 to the plasma membrane is also modulated by phosphorylation, which decreases the membrane association rate.

Phosphorylation may also modulate cluster formation. Clusters formed in all the Pom1 mutant alleles studied here, and exhibited a similar wide range of diffusion rates in Pom1$^{WT}$, Pom1$^{3A}$ and Pom1$^{KD}$, suggesting – if we make the validated assumption that cluster size influences diffusion rate (*Saunders et al., 2012*) – a similar range of cluster sizes. Thus, dephosphorylated Pom1 efficiently forms clusters. Phosphorylated Pom1 may form clusters less efficiently, as Pom1$^{WT}$ forms smaller clusters in tea4Δ cells. We note that it is unclear whether Pom1 is fully phosphorylated in this mutant, or whether it may be inefficiently dephosphorylated by the still present PP1 catalytic subunit lacking the Tea4 regulatory subunit. Previous data had indeed shown that Pom1 dephosphorylation does not strictly require Tea4 (*Kokkoris et al., 2014*). In either case, the data indicate that dephosphorylation at the cell pole also promotes clustering, which may be further favoured by the direct binding with Tea4 (*Hachet et al., 2011*). We note that it may be impossible to fully dissociate cluster formation from membrane binding.

The comparison of Pom1$^{WT}$ and Pom1$^{KD}$ reveals three important differences. First, the diffusion rates of Pom1$^{KD}$ at cell sides were similar to those of Pom1$^{WT}$ at cell poles, consistent with local dephosphorylation of Pom1 at this location. Second, on cell sides the fast pool of Pom1$^{WT}$ molecules was more abundant and diffused faster than Pom1$^{KD}$. This suggests that, by reducing the electrostatic interaction with neighbouring phospholipids, auto-phosphorylation promotes faster mobility of individual Pom1 molecules. Finally, the proportion of single Pom1 molecules increased with distance from cell poles in Pom1$^{WT}$ but not Pom1$^{KD}$. These observations are consistent with progressive phosphorylation-induced dissociation of Pom1 from the membrane and/or clusters.

From these data, we propose a revised model on how the specific gradient shape is achieved. Localized Tea4-PP1 phosphatase activity at cell poles dephosphorylates Pom1, revealing two membrane-binding domains – an amphipathic helix and an adjacent positively charged region – both of which permit the association of Pom1 with the plasma membrane. Pom1 dephosphorylation also favors the formation of clusters at the membrane, which helps carry Pom1 over a longer distance. Within a cluster, individual Pom1 molecules have a short lifetime of 1 to 3 s on average, but new Pom1 molecules join from the cytosol. As clusters split and merge, this permits the transport of single molecules from cluster to cluster. The clusters may diffuse at different rates, either because they are of different sizes and therefore contain different numbers of membrane-binding sites, or because each membrane binding site may be differently phosphorylated and thus bind the membrane with distinct affinity. Indeed, as Pom1 is active on itself, aging clusters become more phosphorylated, which promotes dissociation from the cluster and from the membrane to single molecules. Even though single molecules are present at the membrane a long distance from the poles, their binding is short-lived, detaching from the membrane within 1 s. Thus, clusters act as a diffusion unit, whose regenerating capacity is reduced via progressive auto-phosphorylation with time/distance from the cell pole.

## Pom1 medial cortical levels control mitotic commitment and vary with cell size

One physiological function of Pom1 is to prevent the activation of Cdr2 kinase. Our data clearly establish that the meaningful levels of Pom1 are those at the mid-cell cortex, where Cdr2 forms nodes. Indeed, we see a strong correlation between cell size at division (the phenotypic outcome of Cdr2 regulation) and Pom1 medial levels in both WT and tea4Δ cells. By contrast, the correlations

between cytosolic or cell pole levels of Pom1 and cell length are inconsistent in WT and *tea4Δ* cells, excluding the alternative models that Pom1 may act on Cdr2 at cell poles (*Bhatia et al., 2014*) or in the cytosol. These data agree with the extended size of cells overexpressing Pom1 or mis-targeting it to the medial cortex (*Martin and Berthelot-Grosjean, 2009*; *Moseley et al., 2009*). They also agree with the strong cell lengthening effect of naturally re-distributed Pom1 upon glucose starvation (*Kelkar and Martin, 2015*). Our new observation that *tea4Δ* cells have significant cortical Pom1 signal also fits with the idea that Pom1 remains active on Cdr2 in this mutant as manifested by the longer size of *tea4Δ* than *pom1Δ* (this work and *Martin and Berthelot-Grosjean, 2009*). In this mutant, abundant cytosolic Pom1 likely allows substantial stochastic encounter with, and binding to, the plasma membrane. We note that the phospho-mutant Pom1 alleles did not cause noticeable changes in the position of the division site. This is in agreement with the finding that the two functions of Pom1 in controlling positioning and timing of division are separable (*Bhatia et al., 2014*). This also indicates that a graded Pom1 distribution is less important for division site placement, consistent with observations in *S. japonicus* (*Kinnaer et al., 2019*).

Although Pom1 mid-cell levels clearly correlate with cell size at division in various mutant and environmental conditions, a key unresolved and debated question has been whether Pom1 levels vary during the growth of a single cell, and therefore whether Pom1 may contribute to cell size homeostasis. The higher sensitivity and specificity of TIRF imaging now allows to answer this question unequivocally: there is more Pom1 at the mid-cell cortex of small than long cells. We considered the possibility that this higher concentration may be a simple consequence of cell extension. However, a fission yeast cell doubling in length increases its volume about 1.1-fold more than its surface. Thus, considering a constant Pom1 concentration and an invariant average gradient shape (*Bhatia et al., 2014*; *Saunders et al., 2012*), a higher number of Pom1 molecules have less membrane space at their disposal, which would lead to higher Pom1 concentration at mid-cell in longer cells. Because we observe the opposite, the increased mid-cell levels of Pom1 in short cells must be due to Pom1 gradients extending into mid-cell of small but not long cells. The diffusion rates and cluster lifetimes we have measured are consistent with this scenario. These may be tuned to allow Pom1 diffusion over 3–4 µm distance, sufficient to reach the middle of short, but not long cell, which underlies the difference in Pom1 mid-cell cortical levels in cells of different sizes. Thus, the graded Pom1 distribution is able to convey information in a manner dependent on distance from the source.

These findings poise Pom1 to function as a sensor of cell dimension that provides more inhibition on Cdr2 in short than long cells. This proposition is consistent with biochemical data that Cdr2 activating phosphorylation, which Pom1 counteracts, increases with cell growth (*Deng et al., 2014*) and that Cdr2 is more active in long than short cells, as measured by the number of visits by the Wee1 kinase (*Allard et al., 2018*). It is also consistent with Pom1 function being exquisitely dose-dependent, both in terms of global levels (*Martin and Berthelot-Grosjean, 2009*; *Moseley et al., 2009*) and specifically at the mid-cell cortex (this work). Pom1 likely exerts its inhibitory action primarily on short cells, when its medial concentration is highest. Indeed, blocking Pom1 activity leads to precocious mitosis in short cells lacking other forms of length control, but not in otherwise wildtype cells (*Martin and Berthelot-Grosjean, 2009*). Pom1 may not be the only sensor protein in the pathway: Cdr2 was also proposed to monitor cell surface area through a dynamic exchange of molecules between the cytoplasm and the plasma membrane to form medial nodes whose numbers increase with cell growth (*Facchetti et al., 2019*; *Pan et al., 2014*). Thus, cell growth, by both promoting a decrease in Pom1 levels and an increase in the number of Cdr2 nodes at mid-cell, enhances the activation of Cdr2 in long cells. As mutants in both Pom1 and Cdr2 retain cell size homeostasis, cells likely have secondary sizer mechanisms, perhaps monitoring different geometrical quantities (*Facchetti et al., 2019*; *Wood and Nurse, 2013*).

## Materials and methods

### Yeast strains, media, and genetic approaches

Standard methods of *S. pombe* culturing and genetic manipulations were used. For PALM imaging, *S. pombe* cells were grown in rich yeast extract (YE) medium and imaged during the exponential growth at an $OD_{600}$ comprised between 0.4 and 0.7. All live-cell imagings were performed on YE 2%-agarose pads. For all other imaging experiments strains were grown at 25°C in fully

supplemented synthetic Edinburgh minimal medium (EMM). A complete list of all used strains is provided in *Supplementary file 1*.

For the truncation analysis (*Figure 3D*), Pom1 fragments were amplified by PCR and cloned under the control of the *pom1* promoter in single integration vectors (kind gifts from Dr. Aleksandar Vjestica). The vectors were linearized and integrated at the *ura4* locus in wild type and *pom1Δ* strains. A list of plasmids used in this study is provided in *Supplementary file 2*. The I494N mutation was introduced by site-directed mutagenesis. To generate the MB1* allele, we made use of a native restriction enzyme site (BglII) at aa426 to replace fragment 426–510 with one in which aa429-436 within the conserved region (MB1) were replaced by seven alanines. This mutated fragment was amplified with a forward primer annealing from aa436 onward, and carrying an overhang for the seven alanines and the BglII restriction site. Primer sequences used for mutagenesis are listed in *Supplementary file 3*. To generate *pom1* alleles at the native genomic locus (*Figure 3E*), pSM2142 (containing *pom1* promoter, ORF fused in frame to GFP, kanMX and *pom1* 3'UTR) was used as a backbone for site-directed mutagenesis, except for the *pom1^MB1^\** allele, for which the seven residues were changed to alanine in three rounds of site-directed mutagenesis. To generate the MB1* I494N 5PxxP* mutant, a sequence including the sequence coding for aa81-524 (BspEI to MluI) containing all mutations (as above and in *Hachet et al., 2011*) except PxxP sites 4 and 5 was ordered as a synthetic gBlocks from LubioScience GmbH and introduced in replacement of the wildtype fragment in plasmid pSM2142. PxxP sites 4 and 5 were then introduced by site-directed mutagenesis. The plasmids were digested and transformed into a *pom1Δ::ura4+* background.

The phosphosite mutant allelic series (*Figure 4A*) was generated through site-directed mutagenesis of the sites on a pREP41 plasmid, carrying the full length Pom1 sequence (pSM738). The mutagenized plasmid was digested and transformed in a strain in which *ura4+* replaced the *pom1* sequence coding for aa400-1006 (*pom1Δ(aa400-aa1006)::ura4+*). Integrants were selected on 5-FOA. Each *pom1* allele was subsequently tagged with GFP-kanMX through transformation with linearized pSM1731, a plasmid containing the end of *pom1* ORF without stop codon fused in frame to GFP, the kanMX selection marker and *pom1* 3'UTR. For the Pom1^MB1*-1A(5)^ mutant, phosphosite five was mutagenized through site-directed mutagenesis on pSM2237 (Pom1^MB1*^-GFP) to generate pSM2264, which was digested and transformed into a *pom1Δ::ura4+* strain as above.

All generated strains were verified via sequencing.

Pom1-mEos3.2, Pom1^KD^-mEos3.2, and Pom1^3A^-mEos3.2 strains were generated for PALM microscopy using a standard PCR-based approach (*Bähler et al., 1998*; *Laplante et al., 2016*) and verified by PCR.

BH-search prediction performed at https://hpcwebapps.cit.nih.gov/bhsearch/ with window size for residue averaging of 15 amino acids and values for amino acids set to standard BH parameters.

## Microsized-hole preparation for vertical immobilization of *S. pombe* cells

For cell pole imaging, *S. pombe* cells were vertically immobilized on a YE-2% agarose pad obtained by imprinting on an epoxy resin mold containing an array of micro-pillars (*Wang and Tran, 2014*) with diameter of 6 µm and height of 20 µm (*Figure 1—figure supplement 1*).

SU-8 photolithography and PDMS lithography were performed at the EPFL Center of Microtechnology. SU-8 (GM1060 Gersteltec Sarl) was spin-coated onto a silicon wafer with a thickness of 20 µm and then baked at 95℃ for 40 min. The wafer was then gradually cooled from 95℃ to 30℃ during 30 min. SU-8 polymerization was induced by exposure to 350 nm light for 10 s through a quartz mask containing disk patterns of 6 µm diameter. Post-photolithography baking was performed at 95℃ for 40 min, followed by gradual cooling for 30 min. SU-8 was then developed with manufacturer-provided SU-8 developer and cleaned with isopropanol and dried with compressed air. The SU-8 substrate was hardened by baking at 150℃ for 30 min.

PDMS (Sylgard 184 silicone base, Sylgard curing agent) was then poured onto the surface of the SU-8 mold, which was pre-treated with trimethylchlorosilane vapor (TMCS 33014 from Sigma) to render it non-stick. The PDMS was then polymerized during at least 2 hr at 80℃. The resulting PDMS mold was then used to create the final epoxy resin mold. Epoxy resin R123 bisphenol and epoxy hardener R614 (Soloplast Vosschemie) were poured onto the PDMS substrate and polymerized for 24 hr.

## Super-resolution microscopy for Pom1 dissociation and diffusion dynamics

*S. pombe* cells were genetically modified to express the photoactivable fluorophore mEos3.2 fused to Pom1 protein at the endogenous genomic locus (*Laplante et al., 2016*).

To measure diffusion dynamics, imaging was performed on a previously described custom-built microscope (*Holden et al., 2014*). Cells were imaged in two channels: the fluorescence channel for precise tracking of single Pom1 molecules and the phase contrast channel for cell segmentation and determination of the membrane plane. In order to selectively image Pom1 at the membrane, we used astigmatic imaging to encode the axial position of single molecules (*Huang et al., 2008*). This leads to a change in the shape of the point spread function of molecules not in focus (from circular to elongated), allowing to discard them from the analysis. In a post-processing step, we analyzed single molecules which were at axial positions ranging from −200 nm to +200 nm (with 0 corresponding to the plasma membrane position), eliminating signal detected off the membrane. A z-calibration was performed by taking sequential images of 0.1 µm fluorescent beads (Invitrogen TetraSpeck), displacing the objective by 20 nm steps over an axial range of 1 µm. This calibration allows us to define the PSF widths in the x and y (lateral) directions as a function of the z (axial) displacement. The imaging was performed with an NA 1.49 oil immersion objective lens (Nikon), and fluorescence was detected using an Evolve 128 EMCCD camera (Photometrics) with a 20 ms integration time. Fluorescence was excited with a 560 nm laser (MPB VFL-P-300–560) using an irradiance of 4 kW/ cm$^2$. Molecules of mEos3.2 were photoconverted using a 405 nm laser (Coherent OBIS) with an irradiance of ~0–16 W/ cm$^2$. To measure dissociation, Pom1-mEos3.2 molecules were photoconverted by a pulse of 405 nm and then imaged continuously (no time-lapse +20 ms exposure time) or with the time-lapse sequences (time-lapse durations of 100 + 20 ms or 200 + 20 ms, *Figure 1c*) with 560 nm light until complete bleaching of the photoconverted molecules before the cycle was repeated. Imaging of cells were performed until no more activation of Pom1-mEos3.2 was observed.

Brightfield illumination for phase contrast imaging was performed with a white LED (Thorlabs MCWHL2), passed through a green filter (Chroma ET525/50mc), focused into the back focal plane (BFP) of a condenser lens (Nikon MEL56100). This channel was equipped with a 1024 × 768 CMOS camera (The Imaging Source DMK 31BU03). Image acquisition was controlled through Micromanager.

## PALM localization and image processing

The images were processed with a custom-made ImageJ plugin to segment the single molecule data. Briefly, this consisted of generating a maximum intensity projection (MIP) of all spots, dilating and binarizing the MIP to make a cellular mask and then multiplying this mask by the raw image data. This ensures signal exclusively within cells is analysed and eliminates spurious noise occasionally detected outside of a cell. 3D molecule localization was then performed with RapidSTORM 3.3 (*Wolter et al., 2012*) software, with a z calibration (PSF X and Y width versus Z) used as an input. Single molecules which had an SNR above 50 were localized. Subsequently, a wobble distortion was corrected using an open source Matlab script (*Carlini et al., 2015*); this eliminated lateral shifts in localizations above and below the focal plane. Before tracking, single molecules were filtered based on their integrated intensity and axial position to ensure that only molecules on the membrane were tracked. Localized molecules were then tracked using a MATLAB-based routine based on *Crocker and Grier (1996a)*: molecules belonged to the same track if they were within a 320 nm radius within consecutive frames. No gaps within tracks were permitted. The distance of each track's first point was calculated relative to the cell pole, which was determined from the phase contrast image. First, the phase contrast channel was aligned to the fluorescence channel from images of 500 nm fluorescent beads (Invitrogen TetraSpeck) using a custom-written MATLAB program. Briefly, bead images were localized in 2D and then used to define a rigid transform using a custom MATLAB script. This transformation was then used to map the fluorescence channel onto the phase contrast channel. Second, the center-line of the cell were defined manually using MATLAB's' imline' function, with line extremities corresponding to the poles x,y positions.

## Dissociation and bleaching rate extraction and analysis

The tracking, dissociation and diffusion analysis was performed with custom MATLAB script to extract and study the diffusion and dissociation dynamics.

The recorded molecule trajectories provide an apparent dissociation rate ($k_{eff}$) which is the result of both the molecule photobleaching ($k_{bleach}$) and the actual dissociation rate ($k_{off}$) contributions. By performing time-lapse imaging at time-lapse periods ($\tau_{TL}$) of 20, 120 and 220 ms, we extracted the dissociation constant of Pom1 wide type or mutants.

First, the apparent dissociation rate ($k_{eff}$) is extracted by fitting of the exponential distribution of the track lengths ($t_{eff}$) for every time-lapse experiment (*Figure 1D*; *Figure 1—figure supplement 3A*)

$$f(t) = Ae^{-k_{eff} \times t_{eff}} \tag{1}$$

where the track length is defined as the number of sequential localization events ($n$) spatially separated by less than 321nm multiplied by the time-lapse period $\tau_{TL}$, $t_{eff} = (n-1) \times \tau_{TL}$, and $\tau_{TL}$ is the sum of the camera integration time ($\tau_{int}$– equal to 20ms in our experiment) and the time delays introduced within a pair of two consecutive images.

This distribution is the result of the sum of two independent Poisson processes: the photobleaching that occurs only under laser exposure (i.e. during $\tau_{int}$) and the dissociation of Pom1 that can occur any time during the $\tau_{TL}$ period.

$$k_{eff} \times \tau_{TL} = k_{bleach} \times \tau_{int} + k_{off} \times \tau_{TL} \tag{2}$$

Thus, *Equation 2* can be rewritten as $y(\tau_{TL}) = q + m \times \tau_{TL}$ where $k_{off} = m$ ands $k_{bleach} = q/\tau_{int}$ are easily extracted from the slope and the intercept of the linear fit of $k_{eff} \times \tau_{TL}$ versus $\tau_{TL}$ data. We performed a weighted linear fit where we assigned to each $k_{eff}$ value a weight proportional to the S.D. extracted from the fit of the exponential distribution.

## Data analysis

The emerging of deviation from a linear trend in the log plot of the distribution of the track lengths indicated the possible presence of transient binding with different kinetic constants. Assuming the presence of at least two populations, one with a faster off-rate constant and one with a slower one, the model describing the dissociation of Pom1 then becomes:

$$f(t) = I_{o,a} exp\left(-\left(k_{bleach}\frac{\tau_{int}}{\tau_{tl}} + k_{off,1}\right)t\right) \qquad t_{eff} < TL$$
$$f(t) = I_{o,a}(1-I_{o,b}) exp\left(-\left(k_{bleach}\frac{\tau_{int}}{\tau_{tl}} + k_{off,2}\right)t\right) \quad t_{max} < t_{eff} < TL$$

where $I_{o,a}$ is the number of molecules at start, $t$ is real time in seconds and $I_{o,b}$ represents the percentage of Pom1 molecules exhibiting off-rate constant $k_{off,2}$.

## Acquisition setup

### Fluorescence channel

- Camera: EMCCD Prime 95b
- readout rate = 10 MHz 16-bits
- Exposure time: 20 ms
- Gain: 1000
- Focus locked by CRISP-Focus
- Excitation 560 nm: 'AOTF-DAC1-Wavelength nm'=30 mW
- 'AOTF-DAC1-Volts': '3'
- Activation 405 nm: 'AOTF-DAC3-Wavelength nm'
- 'AOTF-DAC3-Volts'=0 to 5

### Phase-contrast channel

- Camera: TIS-DCAM
- Exposure time: 50 ms

Global fitting parameter results can be found in *Supplementary file 4*.

## Diffusion coefficient and dynamics analysis

Diffusion coefficients were calculated for each track essentially as described in *Manley et al. (2008)*. The individual protein diffusion coefficient (*D*) is extracted from tracks containing more than five consecutive localizations without any gap between localizations using MSD analyser, a MATLAB-based package (*Tarantino et al., 2014*). *D* is derived from the MSD distribution of a Brownian particle's trajectories parameterized through the Einstein–Smoluchowsky equation $MSD = 2dDt$ where *d* is the number of dimensions of the trajectory data (*d* = 2 in this work, since we consider diffusion at the membrane) and *t* is the time lag over which the MSD is measured. *D* can thus be extracted from the slope of linear fit of the first 25% of the mean MSD curve (*Crocker and Grier, 1996b*).

The equation $t = \frac{x^2}{2D}$ was used to derive diffusion distances from diffusion coefficients.

## Microscopy

Confocal microscopy (*Figures 3* and *4*) was performed on an inverted DMI4000B Leica microscope equipped with an HCX Plan Apochromat 100x/1.46 NA oil objective and an UltraVIEW system (Perkin Elmer; including a real-time confocal scanning head CSU22 from Yokagawa Electric Corporation), solid state laser lines, and an electron-multiplying charge-coupled device camera (C9100, Hamamatsu Photonics). Medial section images were obtained at 300 ms exposure time at 100% laser power for five consecutive time points at maximum speed with sum image projections used for quantification and figure preparation. The same settings were used to compare the distributions of Pom1-GFP and Pom1-mEos3.2. For cell length measurements, cells were stained with calcofluor and imaged with a Leica epifluorescence microscope (60X magnification). TIRF microscopy (*Figures 5* and *6*) was performed on a DeltaVision OMX SR imaging system, equipped with a 60 × 1.49 NA TIRF oil objective (oil 1.514), an illumination pathway for ring-TIRF and a front illuminated sCMOS camera size 2560 × 2160 pxl (manufacturer PCO). Imaging settings were: 512 × 512 pxl field of view, 21 ms exposure time, laser power of 20% with TIRF angles: 488 nm at 86.9° and 568 nm 83.2°. Samples were placed on a 0.17 ± 0.01 mm thick glass slide and imaged within 15 min. Imaging was performed in two modes: every second over a 60-s imaging period and every 100 ms over an 8-s imaging period. A widefield image of the medial plane of cells, expressing cytosolic GFP (kind gift from Dr. Magdalena Marek) mixed with a Cdr2-GFP strain (*Figure 5—figure supplement 1C*) was taken as a single snapshot on the same imaging system, using light path setting conventional, 21 ms exposure time, 20% laser power, after which the cortical plane of the same field of view was imaged in TIRF.

## Fluorescence quantifications

Cortical gradient profiles were quantified from confocal microscopy image sum projections by manually drawing a line along the cortex of the cell from the cell tip to the cell middle, generating four gradient profiles per cell in ImageJ (NIH). For the phosphosite mutant allelic series (*Figure 4B*), 240 gradient profiles per strain were generated (n = 3 experiments, 60 cells). 160 gradient profiles were generated for Pom1[MB1*] and Pom1[MB1*-1A(5)] (*Figure 4D*) from n = 2 experiments, 40 cells. The profiles were aligned to gradient maximum intensity value at the cell tip, averaged per strain and plotted against distance from the cell pole. The three individual experiments are shown in *Figure 4—figure supplement 1*. For the quantification of Pom1 intensity levels at cell middle, the average value over the medial-most 1.5 μm of the gradient tails of each individual gradient was calculated and presented as boxplots (*Figure 4C*). We developed a simple MATLAB script to extract decay length (Suppl. Info 1). Briefly, all 240 gradient profiles were aligned to the maximum value at the cell pole and smoothed with a Gaussian filter as previously described (*Hersch et al., 2015*). The profiles were binned by 5% and the first 0.5 μm of the profile values were deleted to avoid the effect of the gradient plateau at the cell pole. The decay length was obtained as the slope of the linear regression on the log of the binned average profiles and plotted against the log of the Pom1 amplitude at the pole. For the correlation plot between cell pole intensities to cell length (*Figure 5—figure supplement 1B*), the intensity at cell pole was calculated as the average values from the first 0.83 μm of individual gradient profiles aligned to the maximum value at cell poles. For the quantification of

cytosolic Pom1 fluorescence intensities (*Figure 4—figure supplement 1C*), the cytosolic signal was quantified per cell from n = 3 independent experiments, total of 50–60 cells. Subtraction of background was performed for both camera noise and auto-fluorescence of a non-GFP-tagged strain that was imaged on the same day for each experiment.

Cell length measurements (*Figure 5A*) were performed on cells from three individual experiments (total number of quantified cells per strain is labeled on the figure), manually drawing a straight line at the detected calcofluor signal from cell tip to cell tip in ImageJ (NIH). Cells imaged at a medial plane. Individual lengths were recorded and presented as boxplots. For correlation plots to Pom1 levels at cell middle or at the cell pole, averages were calculated for each strain.

To quantify Pom1 cluster area and intensity (*Figure 5D*), we drew ROIs around 323 individual clusters from 26 wild type cells and 197 individual clusters from 25 *tea4Δ* cells in ImageJ (NIH). The quantification was done on the image of the first time point from a 60-s imaging period. The cluster area and intensity for phosphomutants Pom1$^{1A}$, Pom1$^{3A}$, and Pom1$^{5A}$ (*Figure 5—figure supplement 2B*) were quantified in the same manner for 37–42 individual cells from three independent experiments. The number of quantified clusters is listed in the figure. For the quantification of cluster duration (*Figure 5G*), raw TIRF images were corrected for bleaching with the EMBL tool Correct-Bleach and further smoothed in ImageJ. Clusters were then tracked individually for the duration of the 8 s imaging period. Fluorescence quantifications from TIRF imaging for global Pom1 levels (*Figure 6G*) were performed by drawing a 45-pixel wide ROI across the entire detectable TIRF signal per cell. To generate two gradient profiles, one from each cell tip, the geometric middle of the cell was used for alignment. To measure intensities at cell middle, the values from the last 1.5 μm of each gradient were averaged. The same method was used for the quantification shown in *Figure 5C*. For quantification of local Pom1 levels at Cdr2 nodes (*Figure 6B*), a nine-pixel ROI was drawn around Cdr2-tdTomato nodes and used to detect signal in the Pom1-GFP channel. Control ROIs of the same size were shifted to the immediate vicinity of a Cdr2 node (*Figure 6C*). Data corrections were done for background camera noise for each individual image and for bleaching. To estimate the average encounter duration, we recorded the length of time the Pom1 signal at a given node was above a defined fluorescence threshold. The threshold choice was instructed by the minimum detectable signal in that experiment. In all cases that threshold was higher than the fluorescence levels measured in the GFP channel using a strain that did not express GFP, in which Cdr2-tdTomato was used to identify the TIRF focal plane and the ROIs of interest. The exact subtracted arbitrary value is indicated in figure legends. Note that the difference observed between short and long cells was robust to changes in threshold choice.

All boxplots plotted using the BoxPlotR web-tool (http://shiny.chemgrid.org/boxplotr/) with definition of whisker extend by Tukey.

## Western blot

Yeast cultures were grown in YE medium at 30°C to OD$_{600}$ = 0.8, collected by centrifugation at 3000r ≤ m at 4°C for 5 min and washed with 1x CXS buffer (50 mM HEPES, pH 7.0, 0,20 mM KCl, 1 mM MgCl2, 2 mM EDTA pH7.5 containing an anti-proteolitic tablet (Roche, Ref 05892791001). Lysates were obtained via mechanical breakage with acid-treated glass beads (Sigma), using a Bead-Beater homogenizer for 10 repetitions at 4.5V of 30 s on, 30 s off on ice cycles. The samples were centrifuged at 10,000 g for 20 min at 4°C and extracts were recovered by pipetting into a new tube. Protein concentration was determined via spectroscopy using Bradford reagent. 270 μg of proteins were loaded per sample on a 10% SDS-PAGE gel and transferred via wet Western blot transfer. 1° antibodies used: α-GFP 1:1000 dilution (Mouse, Roche, Cat.No. 11814460001), α-Tubulin (TAT1, Mouse, 1:5000 dilution), 2° antibodies α-mouse (HRP detection, Promega, W4021). The mean intensity quantification of 4 independent experiments is shown in *Figure 4—figure supplement 1B*. Before averaging, values were corrected for the corresponding background and presented as a ratio of α-GFP to α-tubulin signal.

## Acknowledgements

We thank Dr Marileen Dogterom (TU Delft) and her group for providing the mask for photolithography for the fabrication of the micro-hole, Aster Vanhecke for assistance with processing of some of the tracking data, and Dr Aleksandar Vjestica and Dr Magdalena Marek for gifts of plasmids and

strains. We thank Serge Pelet and members of the Martin lab for critical advice and comments on the manuscript. This work was supported by a Swiss National Science Foundation Sinergia Grant (CRSII3-160728) to SGM and SM.

## Additional information

### Funding

| Funder | Grant reference number | Author |
|---|---|---|
| Swiss National Science Foundation | CRSII3_160728 | Suliana Manley<br>Sophie G Martin |

The funders had no role in study design, data collection and interpretation, or the decision to submit the work for publication.

### Author contributions

Veneta Gerganova, Conceptualization, Investigation, Methodology, Writing—original draft, Writing—review and editing; Charlotte Floderer, Anna Archetti, Lina Carlini, Investigation, Methodology, Writing—review and editing; Laetitia Michon, Thais Reichler, Investigation; Suliana Manley, Conceptualization, Supervision, Funding acquisition, Writing—review and editing; Sophie G Martin, Conceptualization, Supervision, Funding acquisition, Writing—original draft, Project administration, Writing—review and editing

### Author ORCIDs

Veneta Gerganova (iD) https://orcid.org/0000-0002-1491-1249
Anna Archetti (iD) https://orcid.org/0000-0001-9049-3176
Suliana Manley (iD) https://orcid.org/0000-0002-4755-4778
Sophie G Martin (iD) https://orcid.org/0000-0002-5317-2557

### Decision letter and Author response

Decision letter https://doi.org/10.7554/eLife.45983.030
Author response https://doi.org/10.7554/eLife.45983.031

## Additional files

### Supplementary files

• Supplementary file 1. *S. pombe* strains used in this study.
DOI: https://doi.org/10.7554/eLife.45983.024

• Supplementary file 2. Plasmid used in this study.
DOI: https://doi.org/10.7554/eLife.45983.025

• Supplementary file 3. Primers used for mutagenesis. Bold residues indicate changes from the wild-type sequence.
DOI: https://doi.org/10.7554/eLife.45983.026

• Supplementary file 4. Global fitting parameters and outputs for diffusion and dissociation coefficients.
DOI: https://doi.org/10.7554/eLife.45983.027

• Source code 1. MATLAB script for decay length analysis of wild type, Pom1$^{1A}$, Pom1$^{2A}$, and Pom1$^{3A}$ gradients provided with raw data files for each condition.
DOI: https://doi.org/10.7554/eLife.45983.023

• Transparent reporting form
DOI: https://doi.org/10.7554/eLife.45983.028

### Data availability

All data generated during this study are included in the manuscript and supporting files.

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
