## [Decision Letter]

Thank you for submitting your article "Multi-phosphorylation reaction and clustering tune Pom1 gradient mid-cell levels according to cell size" for consideration by *eLife*. Your article has been reviewed by three peer reviewers, and the evaluation has been overseen by Mohan Balasubramanian as the Reviewing Editor and Naama Barkai as the Senior Editor. The reviewers have opted to remain anonymous.

The reviewers have discussed the reviews with one another and the Reviewing Editor has drafted this decision to help you prepare a revised submission.

The reviewers and the reviewing editor appreciated the area under investigation and were enthusiastic about your work on defining the mechanism of regulation of mitotic entry and cell size through the analysis of Pom1 kinase gradient formation and phosphoregulation. In particular, your description of Pom1 behaviour through a variety of advanced imaging approaches was seen as a strength of the paper.

However, a number of major issues have been raised which need to be addressed before this paper can be published in *eLife*.

Essential revisions:

1) This manuscript can be further strengthened if the authors using TIRF microscopy show how the various phospho mutants influence Pom1 cluster size, and duration. In wild type cells the clusters show dynamic changes in intensity over time. Is this dynamicity altered in *pom1* phospho mutants. Does phosphorylation influence cluster formation or cluster stability? These experiments should be easy to perform given that the authors have already shown cluster dynamics using TIRF in WT.

2) The authors predict that Pom1 clusters transiently visit cortical nodes on the plasma membrane to phosphorylate and inhibit Cdr2. One assumption of the model is that most of active Pom1 and Cdr2 are on the plasma membrane, which were not mentioned or tested. If these data are available in the literature, they should be cited. If not, the authors can easily measure the fractions of proteins on the plasma membrane and cytoplasm using fluorescence intensity. Do cytoplasmic concentrations of Pom1 and Cdr2 change with cell size? If most of active Pom1 and Cdr2 are in the cytoplasm, there is a possibility that Pom1 inhibits Cdr2 in the cytoplasm to regulate mitotic entry. If cytosolic Ssp1 can phosphorylate Cdr2, why Pom1 cannot? Is it possible that cytoplasmic Pom1 and Cdr2 are involved in cell-size control directly? To me, it is hard to explain why visits of Weel and Pom1 to cortical Cdr2 are different.

3) Figure 1 shows that Pom1 clusters display different dissociating rates. How are you sure that all the Pom1 clusters are on the plasma membrane? Is it possible some of the clusters are in the cytoplasm? For several experiments, it will be ideal to have a plasma membrane marker so that it is certain that Pom1 is moving or diffusing on the plasma membrane.

4) My primary issue is with the conclusion that *pom1* is regulating cell size. The authors do an excellent job of showing that in cells <10μm that *pom1* is directly inhibiting *cdr2* accumulation (Figure 6). However, for cells >11μm there appears to be little discernible difference between cells in terms of *cdr2/pom1* interaction. Since cells divide with approximately ± 1μm precision, this data does not support the conclusion that *pom1* is regulating cell length at the point of division. Instead, it is consistent with a model whereby *pom1*'s role is to inhibit cell division early on in cell cycle (up to 11um), but after that it is not so vital. For example, Figure 6G has hardly any difference between 9-11 and 12-14μm cells at the cell middle.

So, I think some of the statements regarding *pom1* regulation of cell size need to be toned down – it is acting as a size sensor in early cell cycle, but later other processes are equally, if not more, important.

5) A key prediction from my previous work (Saunders et al., *eLife* 2014) was that the surface area of the cell is the key measure of cell size. This has recently been tested very carefully (Facchetti et al., 2019) and surface area correlates with when division happens much more closely than simply cell length. I find it surprising that the authors do not explore the role of *pom1 / cdr2* interactions in cells of different geometry (e.g. *rga2Δ* and *rga4Δ*). How is *pom1* cluster distribution and *pom1* intensity at the centre altered in cells with different geometry? This is a highly pertinent question for this paper as the problems being addressed are directly related to cell size control.

6) Pom1 and its clusters are, of course, not restricted to the field of view. How did they account for movement of *pom1* molecules/ clusters moving around the surface, rather than disassociating from the surface?

---

## [Author Response]

Essential revisions:1) This manuscript can be further strengthened if the authors using TIRF microscopy show how the various phospho mutants influence Pom1 cluster size, and duration. In wild type cells the clusters show dynamic changes in intensity over time. Is this dynamicity altered in pom1 phospho mutants. Does phosphorylation influence cluster formation or cluster stability? These experiments should be easy to perform given that the authors have already shown cluster dynamics using TIRF in WT.

We have now performed timelapse TIRF imaging of Pom1^1A^, Pom1^3A^ and Pom1^5A^. From these data, we extracted the medial fluorescence intensity, which we now show in Figure 5C (top-right graph). This confirms the data of Figure 4, showing increased intensity of Pom1^1A^, Pom1^3A^ and Pom1^5A^. We also measured cluster area and intensity, now shown in a new Figure 5—figure supplement 2. These measures also show increased size and intensity of clusters in *pom1^3A^*and *pom1^5A^*. We carefully examined the dynamics of these Pom1 phospho-mutant alleles: for all alleles, we could observe events as described in WT, i-e lateral movements, splitting and merging, and oscillatory fluorescence intensity. Though clusters appear more stable in the phospho-mutants, particularly for Pom1^5A^, we have not quantified their lifetime. This is because our measure is already limited by the length of the imaging timeframe in WT (beyond which significant photobleaching compromises interpretation), and so would be non-informative to probe longer lifetimes. Representative timelapse imaging is provided as two new videos (Video 3 and 4).

2) The authors predict that Pom1 clusters transiently visit cortical nodes on the plasma membrane to phosphorylate and inhibit Cdr2. One assumption of the model is that most of active Pom1 and Cdr2 are on the plasma membrane, which were not mentioned or tested. If these data are available in the literature, they should be cited. If not, the authors can easily measure the fractions of proteins on the plasma membrane and cytoplasm using fluorescence intensity. Do cytoplasmic concentrations of Pom1 and Cdr2 change with cell size? If most of active Pom1 and Cdr2 are in the cytoplasm, there is a possibility that Pom1 inhibits Cdr2 in the cytoplasm to regulate mitotic entry. If cytosolic Ssp1 can phosphorylate Cdr2, why Pom1 cannot? Is it possible that cytoplasmic Pom1 and Cdr2 are involved in cell-size control directly? To me, it is hard to explain why visits of Weel and Pom1 to cortical Cdr2 are different.

The analysis of the *pom1* phospho-mutant series in *tea4+* and *tea4∆* backgrounds shows a very clear correlation between the levels of Pom1 at the cortical cell sides and cell size at division. By contrast, new measurements of Pom1 cytosolic levels in these same strains show no correlation with cell size at division: Pom1 cytosolic levels are invariant in the phospho-mutant series in WT and decrease with increasing phosphosite mutations in *tea4∆*, leading to inconsistent relationship with cell length. Thus, the Pom1 medial cortical levels are a much better predictor of cell length at division than the cytosolic levels. Given the direct phosphorylation of Cdr2 by Pom1 (previously shown by Moseley et al., 2009, Martin and Berthelot-Grosjean, 2009, Bhatia et al., 2013, Deng et al., 2014, Rincon et al., 2014), and given cell size at division is a direct read-out of Cdr2 activity, we conclude that this regulation takes place at the cell cortex and not in the cytosol. The new cytosolic measurements and lack of correlation with length at division are provided in Figure 4—figure supplement 1C and Figure 5—figure supplement 1A.

Consistent with the notion that Pom1 regulates Cdr2 at the plasma membrane, Pom1 cytosolic concentration have been indirectly measured in Saunders et al., 2014, and shown not to vary with cell size. We have not re-measured this directly, as we feel that the functional data described above is much more convincing in showing the location of Pom1 action than a likely noisy measurement of a low cytosolic concentration. Our data suggest that visits of Wee1 and Pom1 to Cdr2 nodes are different because Pom1 visits Cdr2 nodes primarily through lateral diffusion in the plasma membrane, whereas Wee1 visits them from the cytosol.

3) Figure 1 shows that Pom1 clusters display different dissociating rates. How are you sure that all the Pom1 clusters are on the plasma membrane? Is it possible some of the clusters are in the cytoplasm? For several experiments, it will be ideal to have a plasma membrane marker so that it is certain that Pom1 is moving or diffusing on the plasma membrane.

There are two reasons we are sure the molecules we tracked were on the plasma membrane and not in the cytosol:

1) Molecules in the cytosol diffuse too rapidly to be localized with high precision at the 20 ms frame rates we used. To image cytosolic diffusion, frame rates of less than 5 ms should be used (for instance as in English BP et al., PNAS (2011)).

2) An astigmatic lens was used to distort the PSF of molecules that were not in focus, and discard them from the analysis. Focusing was readily done into the plane of the plasma membrane, where Pom1 signals were maximum and the shape of the cell was minimized. This is described in the Materials and methods to which we have added explanation of the effect of astigmatic imaging. We also mention it now in the main text, referring to the Materials and methods for details.

4) My primary issue is with the conclusion that pom1 is regulating cell size. The authors do an excellent job of showing that in cells <10μm that pom1 is directly inhibiting cdr2 accumulation (Figure 6). However, for cells >11μm there appears to be little discernible difference between cells in terms of cdr2/pom1 interaction. Since cells divide with approximately ± 1μm precision, this data does not support the conclusion that pom1 is regulating cell length at the point of division. Instead, it is consistent with a model whereby pom1's role is to inhibit cell division early on in cell cycle (up to 11um), but after that it is not so vital. For example, Figure 6G has hardly any difference between 9-11 and 12-14μm cells at the cell middle.So, I think some of the statements regarding pom1 regulation of cell size need to be toned down – it is acting as a size sensor in early cell cycle, but later other processes are equally, if not more, important.

We generally agree with the reviewer and have carefully monitored the wording in the text, changing it where appropriate. It is clear that Pom1 regulates cell size: *pom1∆* cells are shorter than WT. The question is whether it does so by contributing a measure of size that serves as homeostatic control. Because *pom1∆* cells retain homeostatic size regulation, and also because the relative Pom1/Cdr2 levels show significant variability from cell to cell, as does size at division, it is clear that the Pom1/Cdr2 system does not represent the sole mechanism of cell size homeostasis. Our data however show that the variation of Pom1 levels at cell middle according to cell length positions it to be one (of several) size sensors. As the reviewer points out, Pom1 likely contributes more significantly to size control when cells are small than when they have already reached a size over ~11µm. This is entirely consistent with our previous data (Martin and Berthelot-Grosjean, 2009) that Pom1 serves to prevent mitosis specifically in short cells. We have added a couple of sentences at the end of the Discussion clarifying this point.

5) A key prediction from my previous work (Saunders et al. eLife 2014) was that the surface area of the cell is the key measure of cell size. This has recently been tested very carefully (Facchetti et al., 2019) and surface area correlates with when division happens much more closely than simply cell length. I find it surprising that the authors do not explore the role of pom1 / cdr2 interactions in cells of different geometry (e.g. rga2Δ and rga4Δ). How is pom1 cluster distribution and pom1 intensity at the centre altered in cells with different geometry? This is a highly pertinent question for this paper as the problems being addressed are directly related to cell size control.

The finding that cell surface, rather than cell length, is sensed is a very interesting one. Because Pom1 operates at the membrane, it may well monitor cell surface rather than cell length. This would in fact be consistent with the recent proposition of Facchetti et al. that a Cdr2 allele that cannot be phosphorylated by Ssp1 (and is thus also blind to Pom1 regulation) now senses length rather than surface. We have indeed been careful in the text to use the more generic term of “cell size”, which encompasses both possibilities. However, we are not sure at this point how to relate these findings to the behavior of Pom1 clusters and/or relative intensities of Pom1 at cell middle. Finding that there is more cortical mid-cell Pom1 in short cells will be the case whether the sensed dimension is surface of length. Indeed, in preliminary experiments, we found that Pom1 medial cortical levels measured by TIRF imaging are also higher in short than long *rga2∆* and *rga4∆* cells, which are mutants with altered dimensions used in the recent studies. These measurements, performed as in Figure 6G, are provided in Author response image 1. Determining whether and how Pom1 modulates surface measure will require careful quantitative modeling of Pom1 gradient formation in 3D, which goes beyond what we can address in these revisions.

6) Pom1 and its clusters are, of course, not restricted to the field of view. How did they account for movement of pom1 molecules/ clusters moving around the surface, rather than disassociating from the surface?

Please see the response to (3), above.